# A method for measuring the distribution of the shortest telomeres in cells and tissues

Tsung-Po Lai[1], Ning Zhang[1,2], Jungsik Noh[2], Ilgen Mender[1], Enzo Tedone[1], Ejun Huang[1], Woodring E. Wright [1], Gaudenz Danuser[1,2] & Jerry W. Shay[1]

Improved methods to measure the shortest (not just average) telomere lengths (TLs) are needed. We developed Telomere Shortest Length Assay (TeSLA), a technique that detects telomeres from all chromosome ends from <1 kb to 18 kb using small amounts of input DNA. TeSLA improves the specificity and efficiency of TL measurements that is facilitated by user friendly image-processing software to automatically detect and annotate band sizes, calculate average TL, as well as the percent of the shortest telomeres. Compared with other TL measurement methods, TeSLA provides more information about the shortest telomeres. The length of telomeres was measured longitudinally in peripheral blood mononuclear cells during human aging, in tissues during colon cancer progression, in telomere-related diseases such as idiopathic pulmonary fibrosis, as well as in mice and other organisms. The results indicate that TeSLA is a robust method that provides a better understanding of the shortest length of telomeres.

[1] Department of Cell Biology, University of Texas Southwestern Medical Center, 5323 Harry Hines Boulevard, Dallas, TX 75390, USA. [2] Department of Bioinformatics, University of Texas Southwestern Medical Center, 5323 Harry Hines Boulevard, Dallas, TX 75390, USA. Correspondence and requests for materials should be addressed to J.W.S. (email: Jerry.Shay@utsouthwestern.edu)

In vertebrates, telomeres consist of conserved sequence repeats (TTAGGG$_n$) with a single-strand 3′ G-rich overhang. Telomeres reside at the ends of chromosomes and combined with shelterin proteins, help maintain genomic stability[1]. Telomerase is composed of two core components, telomerase RNA (*TR* or *TERC*) and telomerase reverse transcriptase (*TERT*) that add repetitive DNA to telomeres, but is not detected in most human adult somatic cells. Therefore, telomeres progressively shorten with each cell division due to the "end replication problem"[2, 3]. In humans, telomere shortening has been implicated as a risk factor in numerous diseases such as cancer, diabetes mellitus, liver cirrhosis, and cardiovascular disease[4–7]. In addition, genetic diseases

have been identified that have defects in the telomere maintenance machinery termed telomere spectrum disorders (or telomeropathies)[8]. Patients with these syndromes display accelerated telomere attrition and much shorter telomeres compared with age-matched healthy controls[9, 10].

It is well established that it is the shortest telomeres, not average telomere length (TL), that are able to activate DNA damage responses and subsequently trigger a cell-cycle progression arrest, termed cellular senescence[11–14]. Senescence correlates with a variety of age-associated diseases and serves as a tumor suppressor mechanism in large long-lived species to protect genome integrity and prevent accumulation of oncogenic changes

**Fig. 1** Overview of Telomere Shortest Length Assay (TeSLA) and comparison to Universal STELA (U-STELA) and XpYp STELA. **a** Schematic of overall TeSLA methods. Extracted genomic DNA is ligated with TeSLA-Ts (each TeSLA-T contains seven nucleotides of telomeric C-rich repeats at the 3′ end) at the overhangs of telomeres and then digested with a restriction enzyme panel. Digested DNA is subsequently ligated with doubled-stranded TeSLA adapters at the proximal end of telomeres and genomic DNA fragments. After adapter ligation, PCR is performed to amplify ligated telomeric DNA. **b** About 40 pg of DNA from RAJI cells was used in each U-STELA and TeSLA reaction to test specificity of primers for telomere amplification and was tested as indicated (AP, adapter primer; U-TP, U-STELA teltail primer; T-TP, TeSLA-TP). **c** The sensitivity of U-STELA and TeSLA was compared by serial dilution of DNA from RAJI cells from 5 to 40 pg. **d** Using TeSLA (20 pg DNA for each reaction) and XpYp STELA (250 and 500 pg of DNA for each reaction) to detect TL in BJ cells

by limiting the maximum number of cell divisions[15]. Furthermore, an increase in the percent of the shortest telomeres has been proposed to be a predictor of lifespan in mammals[16]. Thus, quantitative information about the shortest telomeres may serve as a biomarker for telomere-associated aging disorders including cancer.

Various approaches have been developed for quantifying TL, and generally provide information on average TL[17–19]. Terminal restriction fragment (TRF) analysis estimates the intensity and size distribution of the "telomeric smear" by Southern blot analysis[20]. The TRF technique requires a large amount of starting genomic DNA, and due to the lower hybridization signal of the shortest telomeres, underestimates the abundance of the shortest telomeres. The quantitative PCR (qPCR) TL measurement[21] has been widely used for high-throughput (HT) testing to overcome the amount of the DNA requirement for TRF analysis and measures ratios of telomere signals (T) to a single copy gene signal (S). However, qPCR only provides relative TL quantitation that is proportional to the average TL from a reference sample (T/S ratio). The qPCR method is not suitable to quantify TL for cancer studies since most cancer cells are aneuploidy[22].

TL can also be measured by quantitative fluorescence in situ hybridization (Q-FISH) methods. Although metaphase Q-FISH[23] can detect TL from each chromosome, it requires dividing cells as well as a skilled cytogeneticist[17, 19]. Flow-FISH is adapted to estimate mean TL of interphase cells (typically human lymphocytes). While this approach is an improvement over qPCR, it requires expensive equipment and the probe not only binds to telomeric repeats but also interacts with non-specific components in the cytoplasm[24, 25]. HT Q-FISH is able to quantify each individual telomere signal in each nucleus, however, telomere clustering has been reported in both lower eukaryotes[26] and mammalian cells[27]. In addition, Q-FISH methods depend on probe hybridization kinetics and do not permit quantitation of the shortest telomeres.

Single telomere length analysis (STELA)[28] was designed to generate high-resolution TL measurements including the shortest telomeres on individual chromosomes. Using ligation and PCR-based methods combined with Southern blot analysis, STELA is able to provide information about the abundance of the shortest telomeres. The major limitation of STELA is that TLs can only be detected on a subset of individual chromosome ends. The Universal STELA (U-STELA)[29] method was reported to solve this problem by detecting telomeres from each chromosome. However, U-STELA is not efficient in detecting TL over 8 kb[29] affecting the detection and accuracy of TL distribution. Computational HT analyses have been developed to evaluate TLs using whole-genome sequencing data[30–32]. The sequencing-based assays only provide average TL and do not correlate well with TRF analysis[33].

Here, we developed a method called Telomere Shortest Length Assay (TeSLA). This method allows more sensitive, efficient, and specific TL detection when directly compared to other methods for TL measurement. We used TeSLA in combination with image-processing software that automatically measures TL after Southern blot analysis. We are thus able to detect telomere dynamics in a range from <1 to ~18 kb in normal aging processes, cancer progression, and telomere-related disorders in humans. Also, we applied the TeSLA method to different mammals such as mice and bowhead whales to demonstrate the utility of this improved technology.

## Results

### The principle of TeSLA.
A schematic presentation of the TeSLA method is shown in Fig. 1a. TeSLA is a ligation and PCR-based

approach to detect amplified TRFs from all chromosomes. TeSLA significantly improves the specificity and the sensitivity for TL measurements when compared to other methods.

First, we compared TeSLA to U-STELA that uses restriction enzyme (RE)-digested genomic DNA to ligate the terminal adapters (telorettes) to the 5′ end of telomeric C-rich strands. However, the U-STELA strategy may result in ligation between subtelomeric sequences and digested genomic DNA fragments. To avoid this possibility and increase the specificity of terminal adapters to anneal and ligate to the 5′ end at each telomeric C-rich strand, the TeSLA method uses extracted DNA (no RE digestion) with a mixture of newly designed terminal adapters (TeSLA-T 1-6; Supplementary Table 1). Each TeSLA-T contains seven nucleotides of telomeric C-rich repeats at the 3′ end, which is complementary to the G-rich overhang followed by a unique sequence derived from bacteriophage MS2 for PCR.

Second, to minimize the measurement of subtelomeric regions and to create specific ends for ligation of adapters at the proximal ends of telomeric repeats, TeSLA uses a combination of four REs (BfaI/CviAII/MseI/NdeI) to digest TeSLA-T-ligated genomic DNA. BfaI and MseI digest DNA at the telomere variant region that is adjacent to the canonical telomere repeats in subtelomeric regions. CviAII and NdeI increase the frequency of generating 5′ AT and 5′ TA overhangs at genomic and subtelomeric regions. We performed TRF analysis using genomic DNA from human BJ fibroblasts and different cancer cells (C106, CEM, HeLa, and RAJI) with REs for TeSLA or two additional different RE mixtures commonly used (AluI/HaeIII/HhaI/HinfI/MspI/RsaI, and HphI/MnlI) that significantly reduce the detection of subtelomeric regions[20, 34]. We observed that the RE mixture for TeSLA further reduces detection of subtelomeric regions when compared to the other combinations of REs (Supplementary Fig. 1a, b). After RE digestion, we performed 5′ dephosphorylation to prevent non-specific ligation between the telomeric DNA fragments and the digested genomic DNA fragments, which could potentially add extra sequences to subtelomeric regions during the next step of TeSLA for adapter ligation.

Third, to increase the ligation efficiency and the specificity of PCR for telomeric DNA amplification, two double-stranded adapters (5′ AT and 5′ TA overhangs) to tag genomic and subtelomeric sequences were generated (Methods; Fig. 1a; Supplementary Table 1). Adapters contain phosphorylated 5′ AT or TA overhang and C3 spacers at each 3′ end to facilitate ligation between the 5′ end of adapters and the 3′ end of genomic/telomeric C-rich DNA fragments. Adapters also contain a unique 3′ overhang complementary to the AP primer for the subsequent PCRs.

Fourth, we used a long-ranged PCR enzyme (FailSafe PCR enzyme mix) that is reliable for telomeric DNA amplification[35, 36] to perform multiple TeSLA PCRs. Because the C-rich telomeric DNA fragments are tagged with TeSLA-Ts and 5′ TA or AT overhang adapters on both ends, multiple copies of tagged telomeres can be amplified using the AP primer together with the TeSLA-TP primer that is identical to the 5′ tail of TeSLA-Ts at each cycle of PCR. Since genomic DNA fragments are ligated only at the 5′ end, the genomic DNA fragments only amplify at one copy per PCR cycle. Amplified DNA fragments are then separated using 0.85% agarose gel and subsequently transferred to positive-charged nylon membrane to perform Southern blot analysis using a hypersensitive telomere-specific probe[37].

To validate the specificity of TeSLA for TL measurement, we used DNA from BJ fibroblasts to perform TeSLA (Supplementary Fig. 2a). In the absence of either RE digestion, adapters, or T4 DNA ligase, no telomere-specific products are detected. We determined the effect of DNA degradation on TeSLA for TL detection using human Jurkat leukemic cells. To obtain Jurkat

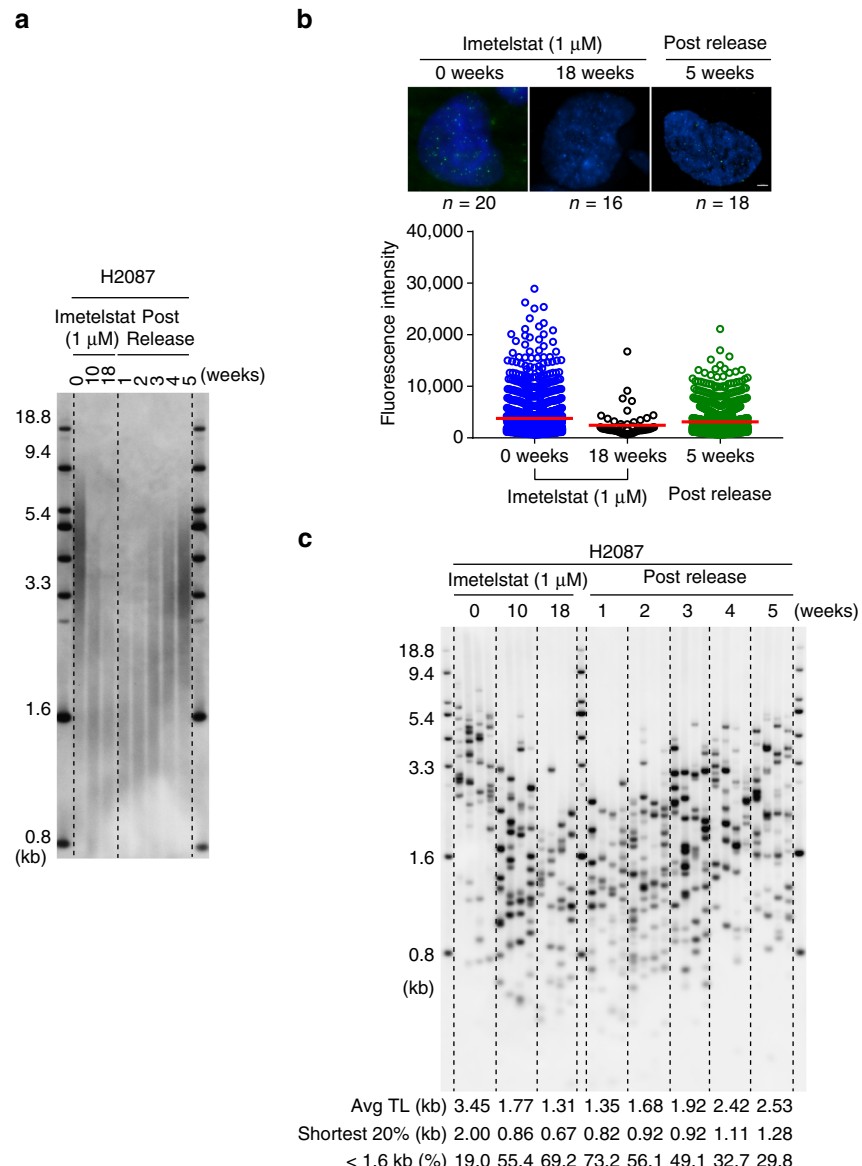

**Fig. 2** Measuring TL in long-term telomerase inhibition with imetelstat (1 μM) treatment and after the removal of drug in H2087 cells. **a** Isolated DNA from H2087 cells with 1 μM imetelstat treatment for 0, 10, and 18 weeks, and post released from 18 weeks for 1, 2, 3, 4, and 5 weeks was digested with the same REs for TeSLA (*BfaI/CviAII/MseI/NdeI*) and then separated on 0.7% agarose gel for TRF analysis. **b** Interphase Q-FISH; cells from H2087 with 0 and 18 weeks 1 μM imetelstat treatment, and 5 weeks after drug removal were used to measure TL by Q-FISH. The results were quantified using TFL-Telo software (*n*: numbers of nuclei were quantified for each time point as indicated above). Scale bar, 3 μm. **c** Results of TeSLA using DNA as indicated. Four TeSLA PCRs (30 pg of each reaction) were performed for each DNA sample

cells with variations in the number of viable cells, we reduced nutrients and allowed the culture medium to become more acidic. We then evaluated the integrity of DNA in cells with different percent (95, 75, 45, and 35%) viable cells and then performed TeSLA. We observed that cells with lower viability have more degraded DNA and a higher frequency of short telomeres (Supplementary Fig. 2b, c). Thus, DNA integrity is essential to obtain reliable results using TeSLA for TL measurement.

**TeSLA compared to STELA and U-STELA.** STELA, U-STELA, and TeSLA are designed to analyze telomere dynamics especially the distribution of the shortest telomeres. Therefore, we compared the sensitivity and specificity of TeSLA to U-STELA and XpYp STELA. In addition to the canonical telomeric ends of chromosomes, short telomere repeats (between 2 and 25 repeats)

called interstitial telomeric sequences (ITSs) are present in numerous intra-chromosomal locations[38, 39]. To preferentially amplify tagged telomeres, the U-STELA uses a "panhandle" design of the proximal adapter to ligate both ends of genomic DNA fragments to suppress subsequent PCR amplification[29]. However, this suppression PCR strategy is designed for low-molecular weight (MW) products[40]. Therefore, the U-STELA method may not completely suppress the amplification of ITSs.

We next compared the specificity of PCR amplification between U-STELA and TeSLA. In the absence of one or both primers, TeSLA does not have any detectable telomere products. However, there are several non-specific PCR products that can be observed after using only one of the primers to amplify the tagged DNA from U-STELA (Fig. 1b). This indicates that the "panhandle" structure may not suppress non-telomeric DNA amplification such as ITSs. In contrast, TeSLA is able to

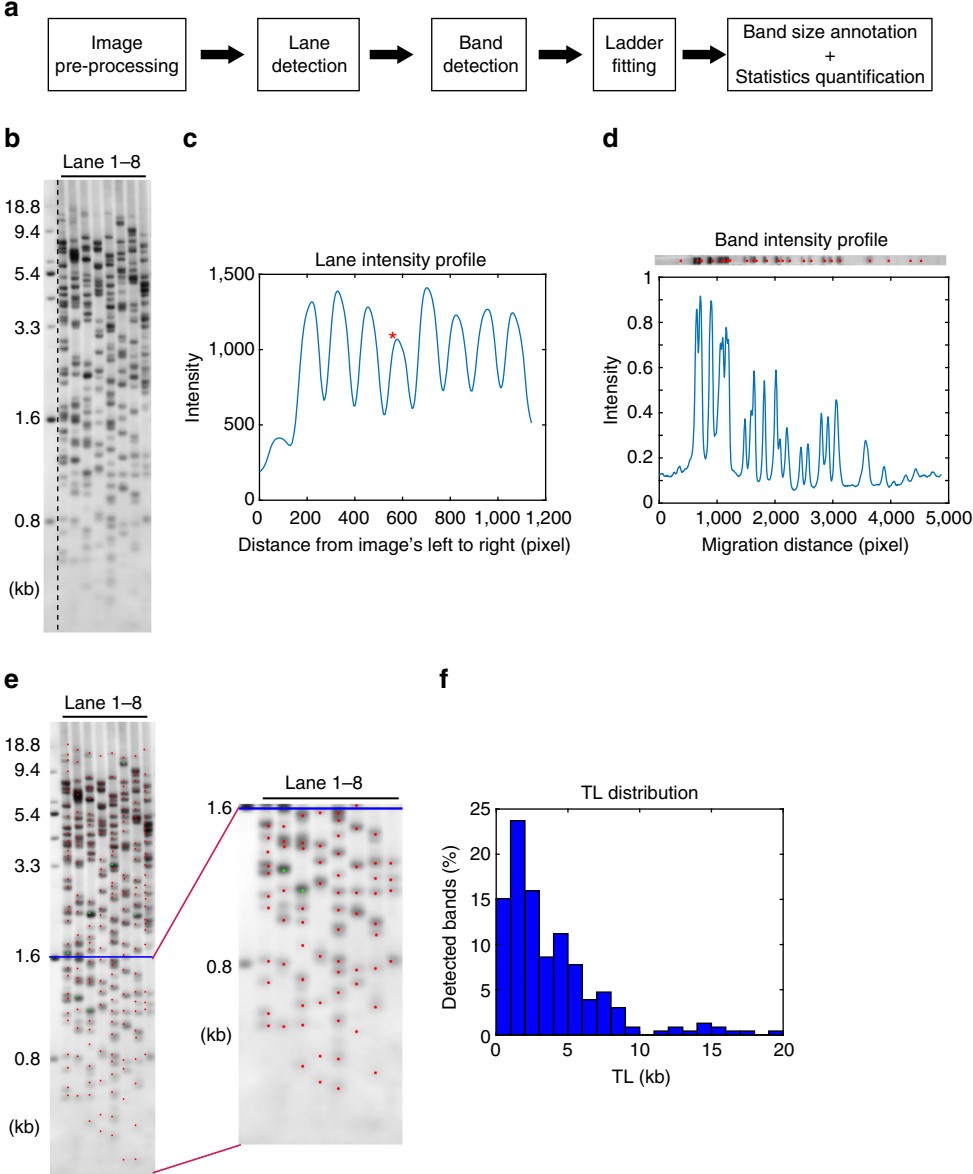

**Fig. 3** Overview of TeSLA image quantification software. **a** The computational analysis pipeline automatically detects each telomere band location, then annotates the band sizes, and calculates the relevant statistics (e.g., average TL, the ratio of shortest TL, TL at 20th percentile, and telomeres below 1.6 kb). **b** Example of input TeSLA image, tiff format recommended. On the left side of the image is the ladder lane with a standard size of 0.8–18.8 kb. **c** Lane profile, generated by summarizing the pixel intensity values vertically from left to right. Each peak indicates one lane detected by the software. **d** Band profile of lane 4, marked with the asterisk in **c**. The band profile is generated for each lane by horizontally summarizing pixel intensity values. Each significant peak refers to an individual band. **e** Example of the final output with the zoom-in of shortest telomere bands. Red dots indicate individual bands. Green dots mark the overlapping bands that are counted twice or three times. The blue line crosses the marker of 1.6 kb, which is the default threshold of shortest TL that other methods cannot reach. The software can calculate the ratio of TL below any given threshold. **f** Histogram of TL distribution, which covers the range of 0–20 kb

specifically detect only chromosomal telomeric repeats without including ITSs (Fig. 1b).

We also performed PCRs on serial dilutions of ligated DNA by TeSLA and U-STELA to compare the sensitivity for telomere detection. Using the same amount of input DNA, we detected more telomere signals using TeSLA compared to U-STELA (Fig. 1c). Thus, TeSLA is more sensitive and efficient compared to U-STELA for telomere detection.

In a normal human cell, each chromosome end has different TLs[14, 23]. Thus, we directly compared the distribution of the telomere amplification products between TeSLA and XpYp STELA using the same BJ fibroblast DNA. The results show

TeSLA is able to identify a wider distribution range of TL in comparison to XpYp STELA using considerably less input DNA (Fig. 1d). When compared to XpYp STELA, TeSLA provides more precise information not only about the average TL but also about all the shortest telomeres, not just those of a specific chromosome end.

**TeSLA is more sensitive compared to TRF and Q-FISH.** To examine the utility of TeSLA in studying telomere dynamics, we evaluated TL in a telomerase positive non-small-cell lung cancer (NSCLC) cell line (H2087). These cells were subjected to long-

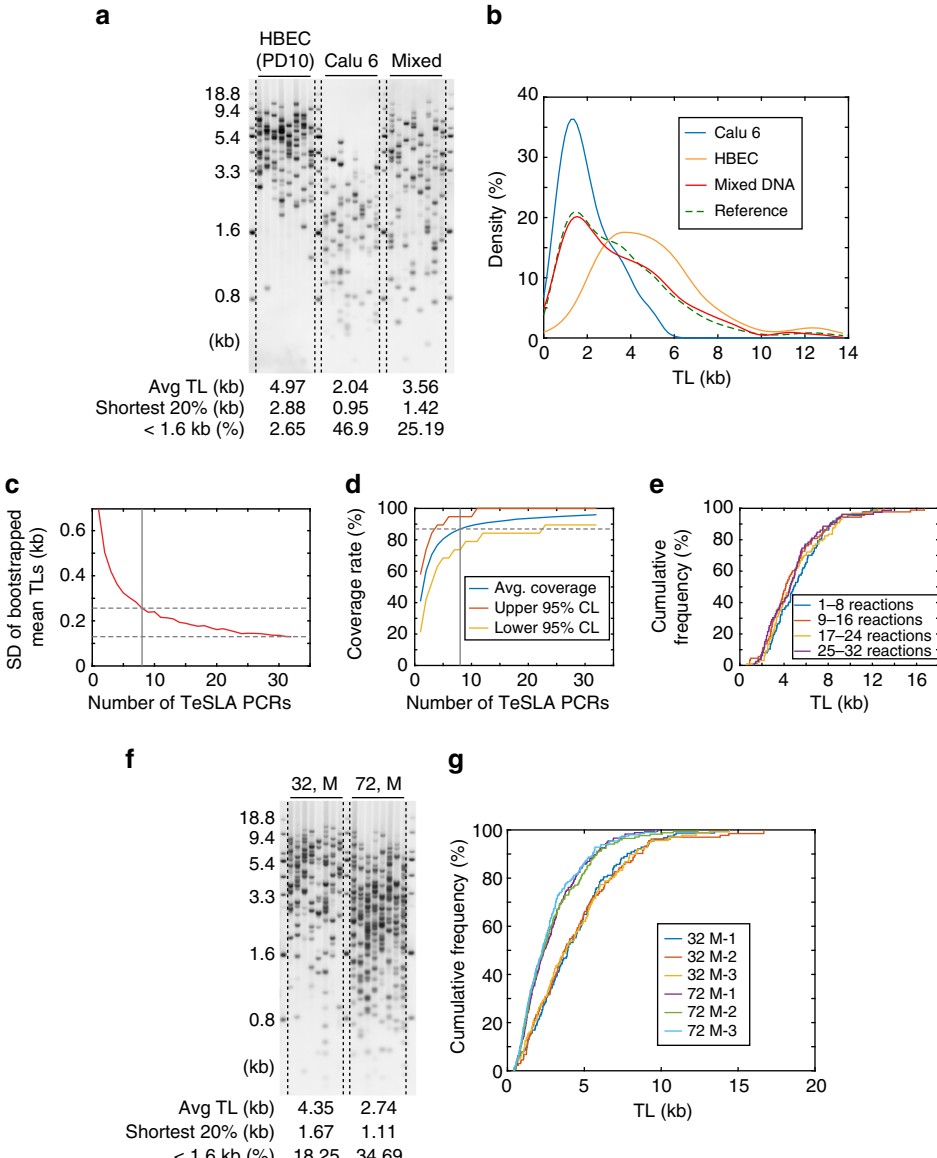

**Fig. 4** Assessment of different variations of TeSLA. **a** TeSLA of human bronchial epithelial cells (HBECs), Calu 6 (lung cancer cell line), and mixed DNA (HBEC: Calu 6 = 1:1) using 30 pg of DNA for each TeSLA PCR. **b** The kernel density estimation of TL from TeSLA results of Calu 6 (blue), HBEC (orange), and mixed DNA (red). The reference line (green dashed) represents a theoretical density function of TeSLA results when HBEC: Calu 6 = 1:1. **c** Standard deviations of bootstrapped distributions of mean TLs computed from 32 TeSLA PCRs of HBECs (Supplementary Fig. 4a) show the estimation accuracy achieved by using $n$ ($1 \leq n \leq 32$) lanes to estimate the mean TL. **d** Coverage rates estimated from 32 TeSLA PCRs of HBECs (Supplementary Fig. 4a) based on bootstrapping. When eight PCRs were randomly selected from the 32 PCRs, 87% of all telomeres were detected with bin sizes 0.5 kb ranging from 0 to 10 kb. Red (yellow) line indicates upper (lower) 95% confidence bounds of the coverage rates. **e** Empirical distribution curves of quadruplets (eight TeSLA PCRs of each) from TeSLA results for HBECs show no significant changes in each eight TeSLA PCRs. **f** Representative TeSLA results of PBMCs from a young (age 32) and an old (age 72) individual. **g** Empirical distribution curves of TLs from the TeSLA of the triplicate results (blue, red, and yellow lines) for the 32-year-old male and triplicate results for the 72-year-old male (purple, green, and sky blue lines)

term telomerase inhibition treatment with imetelstat and then released to observe the dynamics of telomere re-elongation. The results were compared using TeSLA, TRF, and telomere Q-FISH. Imetelstat is a lipid modified thio-phosphoramidate oligonucleotide that binds to the active site of telomerase RNA to inhibit telomerase activity[41]. A previous study[42] demonstrated long-term imetelstat treatment shortened average TL in multiple NSCLC cell lines. In the present study, H2087 cells were continuously treated with 1 μM imetelstat (three times per week) for 18 weeks and then released from imetelstat treatment for 5 weeks. Cells were collected at 10 and 18 weeks with imetelstat treatment and

1–5 weeks after drug removal for TL measurement. Although we observed average TL shortening at 10 and 18 weeks treatment with TRF analyses, the relative intensity of shorter TL measurements, as expected, was significantly reduced (Fig. 2a). We also examined TL using Q-FISH with 0 and 18 weeks of treatment, and 5 weeks in the absence of imetelstat (Fig. 2b). We observed that only a few telomere signals with relatively low intensity were detected in H2087 cells with 18 weeks treatment compared to cells with 0 weeks treatment and 5 weeks after drug removal. This demonstrates that interphase Q-FISH is not sufficiently sensitive to measure TL for cells with extremely short telomeres. In

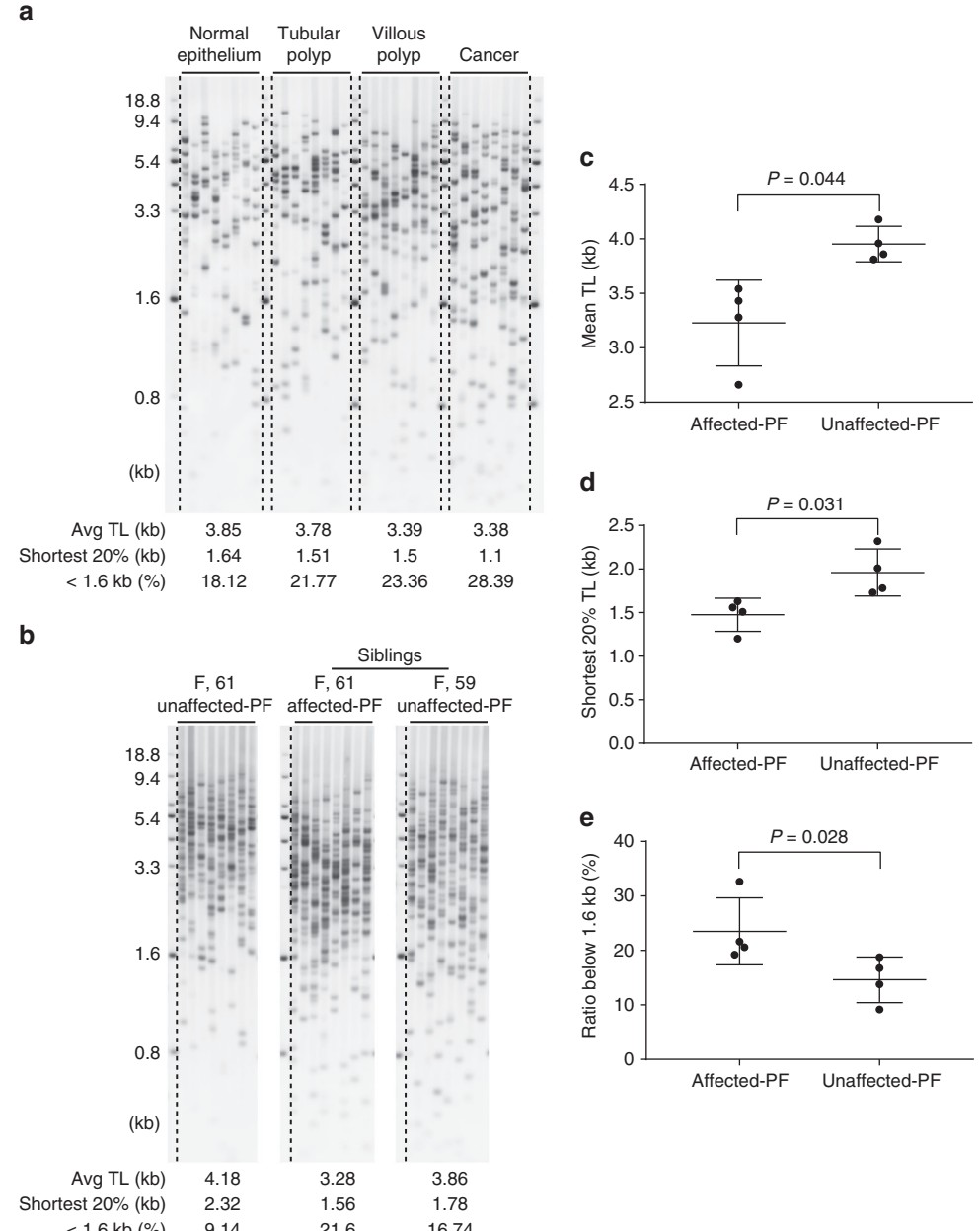

**Fig. 5** Using TeSLA to determine TL and distribution of telomeres in colon cancer progression and idiopathic pulmonary fibrosis (IPF) siblings compared to age-matched normal controls. **a** TeSLA results of normal colorectal epithelium, adenomas (tubular polyp and villous polyp), and colon cancer tissues from one colon cancer patient show shorter mean TL and increasing amount of the shortest telomeres in adenomas and cancer tissues compared to normal colorectal epithelium. **b** Using TeSLA to determine TLs of DNA isolated from circulated leukocytes of the unrelated normal control, siblings with and without IPF. The age and gender are indicated above each TeSLA results. **c**–**e** Scatter plots of mean TL of TeSLA (**c**), the shortest 20% of telomeres (**d**), and percent of the shortest TL (<1.6 kb) (e) are shown for family members that have no IPF (three unrelated controls and a family member without IPF) and four family members with IPF. (**c**–**e** mean and s.e.m., n = 4)

contrast, TeSLA detects TL in both imetelstat-treated cells and cells released from imetelstat in a quantitative manner (Fig. 2c). Using TeSLA, we observed not only the average TL changes but also the changes in the distribution of the shortest telomeres from all chromosomes.

**Software for TeSLA quantification**. To quantify the TeSLA images efficiently and accurately, we developed user friendly software based on MATLAB programming for automatic

detection and size annotation of telomere bands (Supplementary Software). The quantitation workflow is shown in Fig. 3a.

With the pre-processed image (Fig. 3b), the lane profile is generated by summing up the normalized pixel intensity values along each vertical line from left to right (Fig. 3c). The software then detects the center of each lane with watershed segmentation of the lane profile to determine significant peaks. Next, the software estimates the average lane width based on peak-to-peak intervals, and crops the regions of each individual lane for band detection (Fig. 3c). With each individual lane, the software

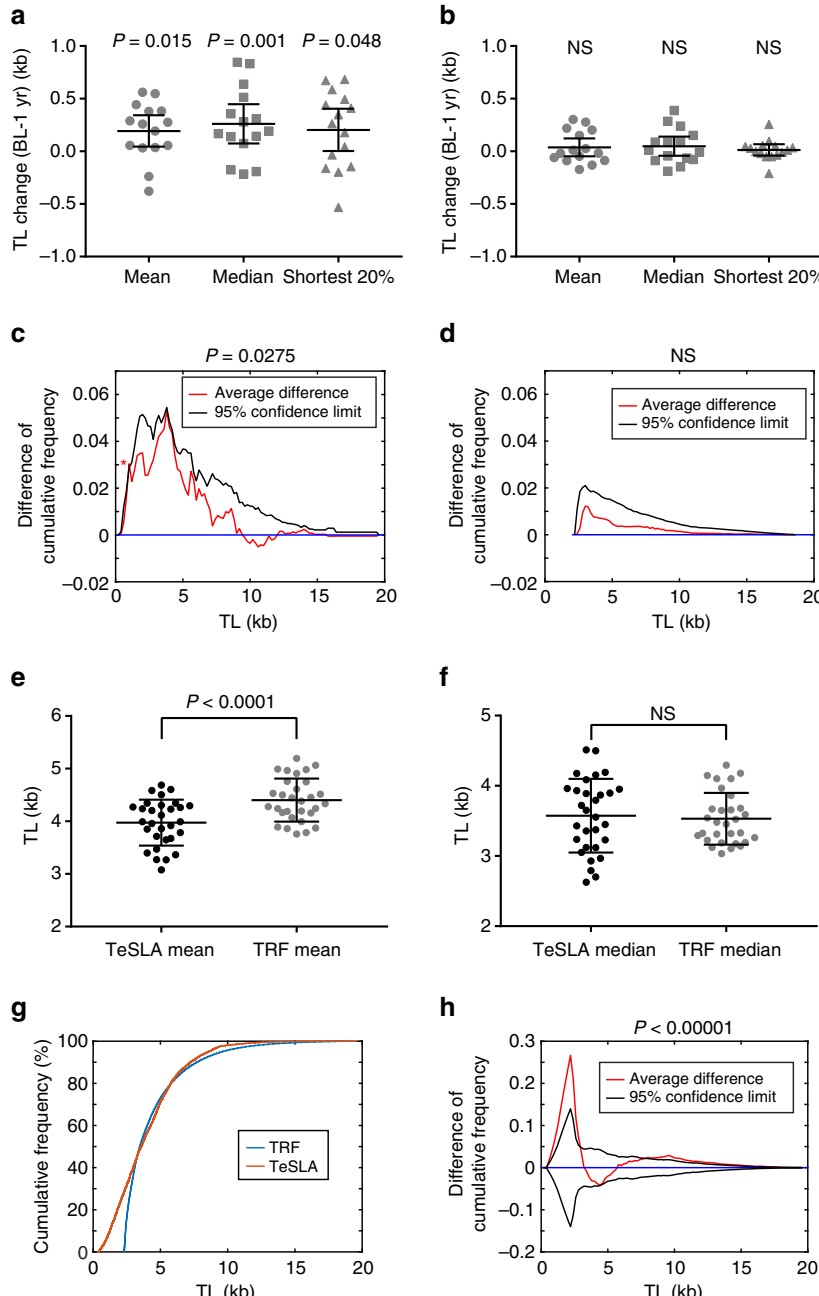

**Fig. 6** TeSLA is sensitive enough to detect changes of TLs in a 1-year period of normal human aging. **a**, **b** Scatter dot blots comparing TLs in PBMCs measured at baseline and in 1-year period by TeSLA (**a**) and TRF analysis (**b**). The mean, the median, and the shortest 20% TLs of 15 normal healthy subjects (age from 51 to 69) were averaged. *P*-values from paired *t*-tests are shown as indicated above. BL, baseline; 1 yr, one year after; NS, not significant. **c**, **d** The average changes of TL distributions in PBMCs in a 1-year period of 15 subjects measured by TeSLA (**c**) and TRF analysis (**d**). One-year differences in cumulative frequencies from each subject were computed (see Supplementary Fig. 5e, f as examples). The average of 1-year changes in TL distributions of 15 subjects is shown in red, and one-sided 95% confidence limit (black) is derived from permutation. The asterisk represents the value (~1 kb of TL) that lies outside the 95% confidence limit, which indicates the most significant effect on telomere shortening. **e**, **f** Scatter plots comparing TeSLA and TRF analysis for mean (**e**) and median (**f**) TL measurements (*n* = 30) in PBMCs. **g** Comparison of TeSLA and TRF analysis of empirical distribution curves of pooled TLs from all 30 DNA samples. **h** The averaged differences (red) in cumulative frequencies (TeSLA-TRF) by the same method used in **c** and **d** show large difference between TeSLA and TRF in the short TL analyses (0.6–2.8 kb). Black lines are 95% confidence limits obtained from permutation. (**a**, **b**; mean and s.e.m., *n* = 15) (**e**, **f**; mean and s.e.m., *n* = 30)

generates band profiles by summing up the pixel intensity values horizontally from top to bottom, followed by segmentation of the significant peaks marking the centers of telomere bands (Fig. 3d). Band intensity is recorded by averaging the pixel intensity values among the local region of bands. The software displays all

detected bands and gives users the ability to manually adjust results.

The software then fits the user-defined ladder size standards at each region of the image. The band size is annotated and recorded by comparing each band to its pixel position and intensity value.

Telomeres with similar sizes that cannot be separated by electrophoresis results in more intense bands. The software is able to identify overlapping bands by comparing the intensity of each band to the intensity of neighboring bands. Significantly brighter bands are attributed with a double or triple count and indicated with different colors in the final output (Fig. 3e). The TL distribution is also plotted for each sample (Fig. 3f). With the annotated band sizes, the software rapidly calculates average TL, the percent of the shortest telomeres (with user-defined threshold), and other relevant statistics.

**TeSLA is not biased for detecting short TLs.** Since TeSLA measures TL using PCR to amplify tagged telomeres, the resolution of TL is limited by the efficiency of PCR amplification for GC-rich telomeric DNA. To examine the upper size limit of TeSLA for TL measurement, we compared TeSLA and TRF on HeLa LT cells, which are telomerase positive and have long telomeres[43]. Using the same panel of REs for TeSLA to perform TRF analysis, we observed that the majority of telomeres distribute at higher MW range (>18.8 kb) with a relatively small amount of telomeres at the lower MW range (<18.8 kb) (Supplementary Fig. 3a). After using ligated HeLa LT DNA to perform 16 TeSLA PCRs, we observed that TeSLA detects TL reliably up to 18 kb (Supplementary Fig. 3b, c). Although we did not use other PCR enzymes to perform TeSLA, we determined that TeSLA's upper limit of telomere detection is ~18 kb, which covers TL detection for the vast majority of human normal and cancer cells.

We next tested whether TeSLA PCR amplification is biased toward amplifying the shortest telomeres. We used DNA from normal human bronchial epithelial cells (HBECs) at early passage (age 24, female), which have relatively long telomeres and a NSCLC cell line, Calu 6, which has very short telomeres[42] to perform TeSLA (Fig. 4a). After ligations, the same amount of ligated DNA from HBEC and Calu 6 was mixed (HBEC:Calu 6 = 1:1) and used for PCR. We observed that the mean TL from the mixed DNA TeSLA (3.56 kb) was very close to the average of HBEC (4.97 kb) in combination with Calu 6 (2.04 kb) results (Fig. 4a). We further visualized the distribution of TL from HBEC, Calu 6, and the mixed DNA using a kernel density estimation (Fig. 4b). The results showed that the distribution of TL from mixed DNA is similar to the distribution of TL from the reference mixture (HBEC:Calu 6 = 1:1) indicating that TeSLA PCR does not have a bias for over-amplifying the shortest telomeres.

**TeSLA for TL measurements.** To achieve representation of all the telomeres in a population of cells, we analyzed the MW of each detected telomere from 32 TeSLA PCRs of HBECs at the same time. Since each reaction detects around 10–15 different MWs (Supplementary Fig. 4a), we determined the number of PCRs that are necessary to obtain a reliable estimate of the entire distribution of TLs in HBECs. We used bootstrapping to draw from the 32 PCRs, many repeats of simulated TL distributions with $1 \leq n \leq 32$ PCRs. We then computed the distribution of mean TLs and corresponding standard deviations (SDs) of the mean TLs to estimate the expected accuracy achievable when "$n$" PCRs were employed to measure the TL (Fig. 4c). When using eight PCRs, the SD was 0.26 kb, about one-third of the SD when using one reaction. Since the gain in estimated accuracy is small with 24 additional reactions, using eight PCRs is reasonable to estimate the mean TL. We also computed coverage rates as a function of the number of reactions used. We observed that eight reactions are sufficient to cover 87% of the bins with a size of 0.5 kb ranging from 0 to 10 kb (Fig. 4d). Based on these analyses, we

used eight PCRs as a unit to analyze the distributions of TL. Next, we divided the 32 PCRs into four data sets and visualized TL distributions from each data set by empirical distribution curves (EDCs) (Fig. 4e) to calculate an intra-assay coefficient of variation (%). For the mean TL, this value is 4.3%.

To determine the inter-variation of TeSLA, we used extracted DNA from human peripheral blood mononuclear cells (PBMCs) provided by two healthy male donor volunteers (age 32 and 72 years old) to perform three independent TeSLAs on different days. We found that PBMCs from the 32-year-old donor have considerably fewer critically short telomeres and longer average TL compared to the 72-year-old donor's PBMCs (Fig. 4f). Then, we visualized the distribution of TL from each triplicate TeSLA result using EDCs (Fig. 4g). The inter-variation between the triplicates was small and the inter-assay coefficient of variations (%) of mean TL was 1.6% for DNA from the 32-year-old donor and 3.9% for DNA from the 72-year-old donor.

To further evaluate whether eight PCRs are sufficient to measure telomere shortening in vitro, we used extracted DNA from two different BJ population doublings (PDs) to perform TeSLA and then quantify TL from both samples (Supplementary Fig. 4b). The results showed that the telomere shortening rate is ~70 bp for each cell division, consistent with previous studies using chromosome-specific STELA and TRF analysis[28, 44].

**TeSLA in cancer and telomere-related disease progression.** Short telomeres correlate with genetic alterations in cancer initiation[45]. To investigate the relationship between cancer progression and telomere dynamics, we performed TeSLA to measure TLs of normal colon epithelium, adenomas (villous and tubular polyps), and colorectal cancer tissues from a patient[46]. We observed using TeSLA that DNA extracted from adenomas and cancer tissues have shorter mean TLs and more of the shortest telomeres below 1.6 kb compared to DNA isolated from normal colon tissue (Fig. 5a).

TRF, Q-FISH, and qPCR methods are widely used to assess TL in telomere spectrum disorders such as idiopathic pulmonary fibrosis (IPF)[47–50]. We used TeSLA to examine leukocyte TLs from eight individuals from a kindred with familial pulmonary fibrosis (four were affected and four were unaffected). We found that the affected family members have more short telomeres and shorter mean TL compared to unaffected family members (Fig. 5b–e).

While the TeSLA results of colon cancer progression and IPF development are consistent with previous studies measuring TL by other methods[46, 49–51], TeSLA provides unprecedented detail of the distribution of telomeres and supports the notion that critically short telomeres may provide information about the stage when short telomeres contribute to cancer initiation and onset of telomere spectrum disorders.

**TeSLA monitors telomere dynamics in normal human aging.** Previous reports indicate that life stresses, infectious diseases, and inflammatory diseases can cause acute telomere shortening[52–54]. We used DNA of PBMCs from healthy subjects who served as placebo-treated volunteers in a clinical trial over a 1-year period[55] to measure TLs by TeSLA and TRF analysis (Supplementary Fig. 5a, b). After comparing changes of TL from each subject over a 1-year period (collected at baseline and 1 year), telomere dynamic changes (15 subjects; 8 females and 7 males, age 51–69) were detected by TeSLA (Fig. 6a) but not by TRF analysis (Fig. 6b).

To further understand the differences between TeSLA and TRF for TL measurements, we used EDCs to represent TL distributions measured by TeSLA and TRF analysis (Supplementary

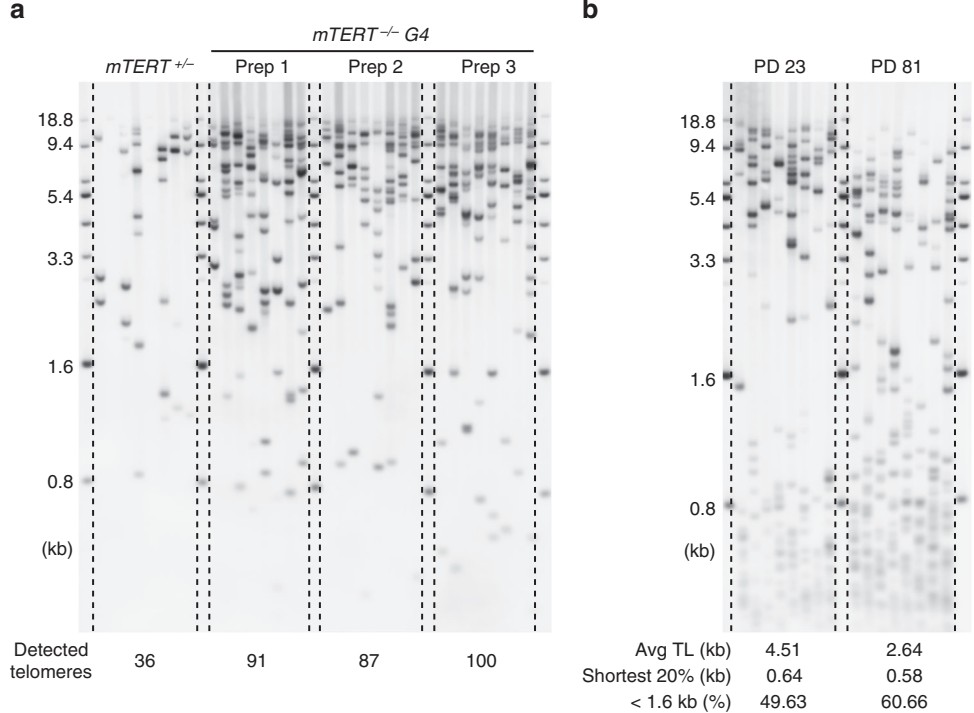

**Fig. 7** TeSLA for telomere detections in *mTERT* knockout mice and lung fibroblasts from a bowhead whale. **a** DNA extracted from *mTERT*+/− and *mTERT*−/− (4th generation, G4) mouse liver tissues were used to perform TeSLA (30 pg for each TeSLA reaction). Detected telomeres of three individual genomic DNA preps from the same *mTERT*−/− G4 mouse (91, 87, and 100 bands) are considerably more than telomeres that were detected from *mTERT*+/− liver tissue (36 bands). **b** TeSLA results of high-quality DNA extracted from early (PD 23) and late (PD 81) passages of cultured bowhead whale lung fibroblasts. Both early and late passage cells contain a subset of the shortest telomeres that have not been identified by TRF analysis

Fig. 5c, d) and then statistically tested changes of TL distributions of each subject in 1 year by comparing differences of cumulative frequency of EDCs from each pair at baseline and after 1 year (Supplementary Fig. 5e, f). Figure 6c, d show the differences of cumulative frequencies averaged (15 pairs) from TeSLA and TRF analysis. These analyses demonstrate that the effect of TL shortening is at relatively short telomeres (<6 kb) in both TeSLA and TRF analysis. Using a permutation-based estimate of the 95% confidence envelope curve, we found that TeSLA not only showed TLs decreased over a 1-year period (one-sided *p*-value 0.0275) but also that the effect of telomere attrition was the most significant in the shortest telomeres (~1 kb) (Fig. 6c). In the TRF analysis (Fig. 6d), although the differences in cumulative frequencies still indicate the effect of TL shortening at short telomeres, the changes in TL distributions during 1 year was not significant (one-sided *p*-value 0.364), consistent with the results of mean, median, and 20th percentile TLs (Fig. 6b).

Next, we directly compared the TL distributions measured by TeSLA and TRF analysis with all 30 DNA samples (15 pairs) at baseline and 1 year. The mean TLs by TeSLA (average 3.97 kb) were consistently 10% shorter than the ones by TRF (average 4.40 kb) (Fig. 6e, paired *t*-test *p*-value < 0.0001). However, the median TLs by TeSLA and TRF analysis were not statistically different (Fig. 6f, *p*-value 0.634). To compare and contrast differences of TeSLA and TRF measurements at the distribution level, we visualized the EDCs from TeSLA and TRF analysis by integrating all 30 TL measurements. The results showed large discrepancies at the shortest telomeres, while the differences of the two distribution curves were moderate at other TLs (Fig. 6g). The averaged difference curve in cumulative frequencies (TeSLA-TRF) together with 95% confidence envelope curves revealed marked differences between TL distributions by TeSLA and TRF analysis in the 0.6–2.8 kb range (Fig. 6h). However, the difference

curve stayed within 95% confidence limits in the range of 3.0–7.8 kb indicating that TeSLA and TRF lead to consistent cumulative frequencies in the middle region and further explains why the median TLs by TeSLA and TRF were not significantly different (Fig. 6f). Thus, TeSLA is able to measure changes of human PBMC TL over 1 year and uncovers distributions of the shortest telomeres that have not been fully addressed by TRF analysis.

**TeSLA for TL measurements in other organisms**. The links between human diseases and TL are studied using tissue samples and cultured cells from humans as well as laboratory mice. Previous studies using Q-FISH analysis demonstrated that progressive telomere shortening causes loss of tissue function in *mTERT*-deficient mice[56, 57]. Most laboratory mice have many telomerase positive tissues with average TLs up to 40 kb[58], which is above TeSLA's upper size limit (~18 kb). However, using TeSLA we tested if we could detect the proportion of the shortest telomeres in mice with TERT deficiency. We extracted DNA from liver tissues of the *mTERT*+/− and the 4th generation *mTERT*−/− mice in a C57BL/6 background. We could detect increasing amounts of telomeres under 18 kb in the 4th generation *mTERT*−/− mouse compared to *mTERT*+/− mouse using the same amount of input DNA (Supplementary Fig. 6a, b) to perform TeSLA (Fig. 7a). In the heterozygous mouse, most of the telomeres are not detected by TeSLA. However, in the *mTERT*−/− mouse (three individual genomic DNA preparations from the same mouse to perform three individual TeSLAs) more short telomeres were detected (Fig. 7a).

DNA sequencing of the longest-living mammal, the bowhead whale, revealed that duplication and loss of genes related to DNA damage responses and repair may be involved in longevity and cancer resistance[59]. The mean TL of cultured lung fibroblast cells

from the bowhead whale is <10 kb by TRF analysis[60]. To examine the telomere dynamics of the bowhead whale in vitro by TeSLA, we used DNA isolated from cultured lung fibroblast cells at different PDs. Even though the DNA from these cells was not degraded, we found that subsets of very short telomeres (<1.6 kb), which have not been reported by TRF or Q-FISH analysis are detected in both early (PD 23) and late passage (PD 81) whale cells. We also observed telomere shortening and increasing amounts of the shortest telomeres at late passage (Fig. 7b).

## Discussion

We developed TeSLA as a method to determine TLs from all chromosomes and to monitor the changes and distribution of the shortest telomeres, the average TL, as well as TLs up to 18 kb. By directly comparing TeSLA to U-STELA and XpYp STELA for TL measurements (Fig. 1), we demonstrated that TeSLA is more sensitive and specific for TL detection and generates more information of the spectrum of telomere distributions. We also compared TeSLA to both TRF and Q-FISH methods for detecting extremely short telomeres after treating a NSCLC cell line with a telomerase inhibitor (imetelstat) and then after imetelstat was removed (Fig. 2). With TeSLA, but not TRF analysis or Q-FISH, we were able to quantitate the length of the shortest telomeres. In addition, we developed software to automatically quantify TeSLA results (Fig. 3).

We demonstrated that TeSLA is able to measure telomeres up to 18 kb reliably (Supplementary Fig. 3). Although TeSLA is sufficient to detect TL in almost all human normal somatic and cancer cells, TeSLA might not be suitable for cells, which have long and heterogeneous TLs such as alternative lengthening of telomeres (ALT) cells[61]. In addition, we demonstrated that TeSLA is not biased for amplifying the shortest telomeres and is a highly reproducible method for TL measurements (Fig. 4). Together, these analyses document that TeSLA measures the TL distribution in cells with a higher degree of confidence compared to existing TL measurement approaches.

To illustrate the changes of TL in human diseases, we examined telomere dynamics in a colon cancer progression series and a familial kindred with IPF (Fig. 5). With TeSLA, we observed not only short mean TL but also increasing amounts of very short telomeres that correlate with cancer and IPF progression. Going forward, TeSLA may serve as a tool to detect telomere shortening of clinical disease onset.

With TeSLA, but not TRF analysis, we detected changes in TL of PBMCs from healthy subjects over a 1-year period (Fig. 6). Others have demonstrated the complexity of TLs measured in PBMCs by identifying variations in TLs and the rates of TL changes that are cell type-specific in vivo[62, 63]. TeSLA is very sensitive for detecting the shortest telomeres in a heterogeneous telomere background. Thus, TeSLA is capable of measuring sub-populations of cells in PBMCs, such as CD28- T cells, that have a lower capacity for cell division, shorter telomeres, and higher TL shortening rate compared to other sub-types of cells in PBMCs[64]. Thus, TeSLA may be able to identify critically short telomeres in specific subsets of immune cells that are important in human aging. Using newly designed statistical analysis tools to compare TL distribution over a 1-year period, we were able to determine the most dramatic effect on telomere shortening is on some of the shortest telomeres (around 1 kb in length) from a group of healthy human volunteers. This suggests that TeSLA could be used to detect pathological thresholds of disease at an earlier stage than previously possible. Early diagnoses may result in the implementation of more effective interventions.

Besides studying changes of TLs in humans, TeSLA can be applied to evaluate the shortest telomeres in other animals (Fig. 7). We demonstrated that TeSLA detects the distribution of the shortest telomeres in *mTERT* knockout mice. A recent study reported that telomere shortening is a critical factor for age-dependent cardiac disease in the NOTCH1 haploinsufficiency mouse model[65]. Thus, while TeSLA may not be able to detect the longest telomeres in mice, it can serve as a powerful tool to study the relationship between changes of the shortest telomeres and age-dependent diseases in mouse models with deficiencies in telomere maintenance.

TeSLA provides resolutions of all the telomeres including the shortest and requires very small amounts of starting DNA without limitation of specific cell types or tissue samples. TeSLA, therefore, will be useful for studying the changes of the shortest telomeres in disease development (pathological thresholds). Finally, TeSLA can provide information about telomere dynamics in human cells as well as other animals.

## Methods

**Cell culture.** BJ, Calu 6, HeLa, HeLa LT, HT1080, and NIH 3T3 cells were maintained in Media-X with 10% cosmic calf serum (Hyclone) at 37 °C in 5% CO$_2$. C106, CEM, Jurkat, and RAJI cells were cultured in RPMI with 10% of fetal bovine serum at 37 °C in 5% CO$_2$. H2087 cells were cultured at 37 °C in 5% CO$_2$ in Media-X containing 10% cosmic calf serum with or without 1 μM imetelstat. HBECs were cultured in bronchial epithelial growth medium (BEGM) (Lonza, Allendale, NJ) with an antibiotic solution (penicillin G-streptomycin–amphotericin B) and incubated in low oxygen (2–3%) at 37 °C. Bowhead whale lung fibroblasts were maintained in F12/DMEM (1/1) with 15% of fetal bovine serum and incubated in low oxygen (2–3%) on 0.1% gelatin-coated flasks at 32 °C. All cells were tested regularly for mycoplasma contamination.

**TRF analysis.** Isolated DNA (3 μg of each sample) was digested with different combinations of REs (as described in the figures) and then separated on a 0.7% agarose gel at 2 V/cm for 16 h. The telomere signals were detected by Southern blot analysis as previously indicated[37]. In brief, after DNA was transferred from gel to a positive-charged nylon membrane, the transferred DNA was fixed by UV cross-linking. The cross-linked membrane was then hybridized with the hypersensitive DIG-labeled telomere probe overnight at 42 °C. After hybridization, the membrane was washed with buffer 1 (2× SSC, 0.1% SDS) at room temperature for 15 min and then washed twice with buffer 2 (0.5× SSC, 0.1% SDS) at 55 °C for 15 min. Next, the membrane was washed with DIG wash buffer (1× maleic acid buffer with 0.3% Tween-20) for 5 min. Then the membrane was incubated with 1× DIG blocking solution for 30 min at room temperature. After blocking, the membrane was incubated with anti-DIG antibody (Roche) in 1× blocking solution (1 to 10,000 dilution) for 30 min at room temperature. The membrane was then washed with DIG buffer two times at room temperature for 15 min. After washing, telomere signals were detected by incubating with CDP-star for 5 min. Image quantification was performed using Image Quant software to measure the intensity value of each telomere smear. The intensity value of each sample was then adjusted with the background graphed from a lane with no DNA sample[66].

**Quantitative florescence in situ hybridization.** Cells in interphase were fixed with freshly prepared methanol/acetic acid (3/1 vol/vol) and then were stored at −20 °C. Fixed cells were dropped onto methanol/acetic acid pre-wet slides and subsequently air-dried overnight. We used the Q-FISH protocol as previously described[67]. In brief, slides with fixed cells were washed three times with PBS for 5 min and then with PBS containing 0.5% Triton X-100 for 10 min. Fixed cells were washed three times with PBS for 5 min and then dehydrated with 70, 90, and 100% ethanol for 5 min each. Cells were hybridized with FAM-TelC probe (panagene) in hybridization buffer (70% foramide, 0.6× SSC, 5% MgCl$_2$, and 0.0025% blocking reagent (Roche)) at 80 °C for 7 min and then incubated at room temperature for 12 h. After incubation, slides were washed with wash buffer (70% foramide and 0.6× SSC) for 1 h and 2× SSC for 30 min. Cells were washed with PBS for 5 min and then dehydrated with 70, 90, and 100% ethanol for 5 min each. Slides were mounted with VECTASHIELD Antifade Mounting Medium with DAPI (Vector Labs). Imaging was acquired using a Personal DeltaVision wide-field fluorescent microscope[68]. Image quantification was performed using TFL-Telo V2 image analysis software[69].

**Genomic DNA extraction.** Genomic DNA was extracted using the Gentra Puregen DNA Extraction Kit (Qiagen) according to the manufacturer's instructions. Each DNA sample was evaluated on a Nanodrop (Thermo Scientific) for concentration and purity, and integrity of DNA was determined by resolving 20 ng of DNA on a 1% agarose gel[20].

**XpYp STELA and Universal STELA**. We used the XpYp STELA protocol as previously indicated[35] with some modifications. In brief, 10 ng of EcoRI digested DNA was ligated with a mixture of C-telorettes 1–6 ($10^{-3}$ μM in 10 μl reaction at 35 °C for 12 h. About 250–500 pg of ligated DNA was used to perform multiple PCRs with 0.1 μM of XpYpE2 and Teltail primers (26 cycles of 95 °C for 15 s, 58 °C for 20 s, and 72 °C for 15 min). The PCR products were resolved on a 0.85 % agarose gel (1.5 V/cm for 19 h). The Southern blot analysis was performed as described for TRF analysis to detect amplified telomeres.

For the Universal STELA, we used the protocol as previously described[29]. In brief, 10 ng of MseI and NdeI digested DNA with 50 μM 42-mer and 11 + 2-mer oligonucleotides were incubated at 65 °C and then gradually cooled down to 16 °C for 1 h. The mixture was then incubated at 16 °C with T4 DNA ligase for 12 h in 10 μl reaction. The mixture was incubated with $10^{-3}$ μM of C-telorettes (as indicated for XpYp STELA) in 25 μl ligation reaction at 35 °C for 12 h. About 5–40 pg of ligated DNA was used to perform multiple PCRs with 0.1 μM of Adapter and Teltail primers
(1 cycle at 68 °C for 5 min, 26 cycles at 95 °C for 15 s, 58 °C for 20 s, and 72 °C for 15 min). The PCR products were separated by a 0.85% agarose gel. Telomere signals were detected by Southern blot analysis as described for TRF analysis.

Oligonucleotides for XpYp STELA and Universal STELA:
C telorette1, 5′-TGCTCCGTGCATCTGGCATCCCCTAAC-3′;
C telorette 2, 5′-TGCTCCGTGCATCTGGCATCTAACCCT-3′;
C telorette 3, 5′-TGCTCCGTGCATCTGGCATCCCTAACC-3′;
C telorette 4, 5′-TGCTCCGTGCATCTGGCATCCTAACCC-3′;
C telorette 5, 5′-TGCTCCGTGCATCTGGCATCAACCCTA-3′;
C telorette 6, 5′-TGCTCCGTGCATCTGGCATCACCCTAA-3′;
XpYpE2, 5′-TTGTCTCAGGGTCCTAGTG-3′;
Teltail, 5′-TGCTCCGTGCATCTGGCATC-3′;
Adapter, 5′-TGTAGCGTGAAGACGACAGAA-3′;
11 + 2-mer, 5′-TACCCGCGTCCGC-3′;
42-mer, 5′-
TGTAGCGTGAAGACGACAGAAAGGGCGTGGTGCGGACGCGGG-3′.

**TeSLA for TL measurement**. Before starting the TeSLA procedure, we make stocks of short double-stranded 5′ AT and TA overhang adapters for ligation at genomic and subtelomeric regions. To make 40 μM AT and TA adapters, 40 μl of 100 μM TeSLA adapter short oligonucleotide (ONT) was mixed with 40 μl of 100 μM of TeSLA adapter TA and TeSLA adapter AT ONTs individually to make the final volume of 100 μl in 1× TSE buffer (10 mM Tris pH 8.0, 50 mM NaCl, 1 mM EDTA). The mixtures were incubated at 95 °C for 5 min and subsequently allowed to gradually cool down to room temperature.

To ligate TeSLA-T (TeSLA-T 1–6) to each telomere overhang, 1000 units of T4 DNA ligase (New England Biolabs), 1 mM ATP, and $10^{-3}$ μM of TeSLA-Ts were added to a final volume of 20 μl in 1× CutSmart buffer (New England Biolabs) with 50 ng of isolated genomic DNA (without RE digestion). The mixture was incubated at 35 °C for 12–16 h and then heat inactivated at 65 °C for 10 min. The inactivated mixture including two units of CviAII in 10 μl 1× CutSmart buffer was incubated at 25 °C for 2 h to generate genomic DNA fragments with 5′ AT overhangs. After CviAII digestion, two units of BfaI, NdeI, and MseI (New England Biolabs) in 10 μl 1× CutSmart buffer was added to the CviAII digested mixture to further digest genomic DNA as well as to generate DNA fragments with a 5′ TA overhang by incubating at 37 °C for 2 h. One unit of Shrimp Alkaline Phosphatase (rSAP; New England Biolabs) in 10 μl 1× CutSmart buffer was subsequently added to the digested mixture to remove 5′ phosphate from each DNA fragment to improve the specificity of ligation between overhang adapters and genomic DNA fragments (37 °C for 30–60 min) and subsequently heat inactivated at 80 °C for 20 min. After heat inactivation, the mixture was allowed to gradually cool down to 25 °C.

For adaptor ligation, 10 μl of the inactivated mixture was combined with 1000 units of T4 DNA ligase to a final volume of 20 μl in 1× CutSmart buffer with 1 mM ATP, 1 μM of AT adapter, and 1 μM of TA adapter. The mixture was incubated at 16 °C for 12–16 h. After adapter ligation, the mixture was heat inactivated at 65 °C for 10 min.

After adapter ligation, multiple PCR reactions were performed (initial melt at 94 °C for 2 min followed by 26 cycles at 94 °C for 15 s, 60 °C for 30 s, and 72 °C for 15 min) using 2.5 units of FailSafe Enzyme Mix (Epicenter) with 1× FailSafe buffer H in 25 μl reaction containing 0.25 μM primers (AP and TeSLA-TP) and 20–40 pg of ligated DNA. PCR products were resolved on a 0.85 % agarose gel (1.5 V/cm for 19 h). After gel electrophoresis, the Southern blot analysis was performed as described for TRF analysis to detect amplified telomeres.

**Image quantification and statistical analysis**. TeSLA is able to provide information for shortest TL <3 kb, which is often missed in other TL quantification methods. For this assay, we developed user friendly software that can automatically detect bands from sample images, annotate band size, calculate average TL, the percent of the shortest telomeres (1.6 kb and shorter), and obtain other relevant statistics. We have documented the validity of this assay in a spectrum of samples. The executable file, Matlab code, and detailed step-by-step manual of this software are attached in the Supplementary Item.

**Bootstrapping method to assess estimation accuracy**. We applied a bootstrapping method to 32 TeSLA PCRs of HBEC cells to evaluate the estimation accuracy that would be achieved by using $n$ reactions ($1 \leq n \leq 32$) so that we could determine the number of reactions that give reasonably stable TL measurements. We computed the variation of mean TLs as well as the coverage rates to determine whether the detected TLs are sufficiently dense within 0–10 kb. In the bootstrapping method, we randomly selected $n$ reactions with replacement, computed the mean TL, and repeated this procedure 500 times for each number of reactions. We then computed SDs of the bootstrapped mean TLs, which showed the estimation accuracy as a function of the number of reactions (Fig. 4c). To compute coverage rates, we considered bins with a size of 0.5 kb ranging from 0 to 1. Coverage rates at $n$ reactions were defined to be the percentage of bins containing at least one reaction when randomly chosen $n$ reactions are used. The coverage rate curve together with confidence bounds estimated from variation of bootstrapping repeats is shown in Fig. 4d.

**Permutation test for comparison of paired EDCs**. We developed a new statistical test based on permutation for pairwise comparison of two sets of TL distribution curves to perform a systematic analysis of TL changes between two experimental conditions along the entire length spectrum or in a particular, yet a priori unknown sub-spectrum. This test was applied, e.g., to compare TeSLA measurements for PBMCs from 15 subjects measured at baseline and 1 year after (Fig. 6c). Without loss of generality, we explain the testing procedure for this data set. We first represented TL measurements at baseline and 1 year after with EDCs for each subject (Supplementary Fig. 5c, d), and computed 1-year differences in cumulative frequencies at every value of TLs (Supplementary Fig. 5e, f). The difference curves were averaged over subjects. The resulting averaged difference curve (Fig. 6c, d) visualizes 1-year changes in TL distributions.

To test how much the observed averaged difference curve deviates from zero, i.e., the null hypothesis that TL distributions do not change over 1 year, we randomly permuted for each subject independently the temporal order of the experimental condition, i.e., baseline and 1 year after. Accordingly, 50% of subjects, on average, retained the original order, whereas 50% of subjects had the order reversed. This procedure was repeated $10^5$ times to simulate permuted averaged difference curve under the reference situation without difference between the conditions. Next, we derived 95% confidence envelope curves about the reference situation (Fig. 6c, d). If the experimentally observed averaged difference curve goes outside the envelope curve at any TL value, then the null hypothesis is rejected with a $p$-value < 5%, i.e., the distributions did likely change between the two compared conditions.

To derive the one-sided 95% confidence envelope curve, we considered the EDCs measured at baseline and in 1-year period ($F_{0i}(x)$, $F_{1i}(x)$), for all subjects $i = 1, 2, \ldots, n$, with $x > 0$ denoting a TL. One-year changes at the distribution level can be visualized by the observed averaged difference curve

$$\overline{D}(x) = \frac{1}{n} \sum_{i=1}^{n} \left\{ F_{1i}(x) - F_{0i}(x) \right\}. \text{ Let } \overline{D}^{(k)}(x) \text{ denote the averaged difference curve}$$

from $k$-th permuted data set. These permuted difference curves displayed heterogeneous variance at different values of TLs, which is intrinsic to the data structure. Before quantifying deviation of the observed difference curve from the reference difference curve, the permuted difference curves are standardized to account for the heterogeneous variance as follows:

$$Z^{(k)}(x) = \frac{\overline{D}^{(k)}(x) - m_{\text{perm}}(x)}{\sigma_{\text{perm}}(x)}, \, k = 1, 2, \ldots, K,$$

where

$$m_{\text{perm}}(x) = \frac{1}{K} \sum_{k=1}^{K} \overline{D}^{(k)}(x), \, \sigma_{\text{perm}}(x) = \left\{ \frac{1}{K-1} \sum_{k=1}^{K} \left( \overline{D}^{(k)}(x) - m_{\text{perm}}(x) \right)^2 \right\}^{1/2},$$

and $K$ is the total number of permutations. We then obtained permutation distances of standardized difference curves from the reference function in positive direction by $T_{\text{perm}}^{(k)} = \max_{x>0} Z^{(k)}(x)$. The distance between the observed and the reference difference curves is computed by $T_{\text{obs}} = \max_{x>0} \left\{ \left( \overline{D}(x) - m_{\text{perm}}(x) \right) / \sigma_{\text{perm}}(x) \right\}$ using the same standardization. The significance of the observed distance can now be assessed by comparing it with the permutation distances and computing the $p$-value,

$$P_{\text{perm}} = \#\left\{ T_{\text{perm}}^{(k)} > T_{\text{obs}} \right\} / K.$$

The one-sided 95% confidence envelope curve is obtained by $Q(0.95) \times \sigma_{\text{perm}}(x) + m_{\text{perm}}(x)$, where $Q(0.95)$ is the 0.95th quantile of the permutation distances.

**Quantitative PCR**. The relative amount of DNA from $mTERT^{+/-}$ mouse liver tissue and three individual DNA preps from the same $mTERT^{-/-}$ mouse liver

tissue was measured by qPCR (initial melt at 95 °C for 3 min followed by 35 cycles at 95 °C for 10 s, 60 °C for 1 min) using SsoFast EvaGreen Supermix (Bio-Rad) in a LightCycler 480 II (Roche Molecular Biochemicals). Relative DNA levels of each mouse DNA were calculated and normalized to the mean of amplification level of mouse B1 repeats from 10 ng of genomic DNA from NIH 3T3 cells. Primers for mouse B1 repeats are listed in Supplementary Table 1.

**Data availability**. The authors declare that all the relevant data supporting the findings of this study are available within the article and its supplementary information files, as well from the authors upon reasonable request.

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

## Acknowledgements

We thank Dr. Lea Harrington from Université de Montréal for providing the mTERT knockout mice. We also thank Dr. Christine K. Garcia from University of Texas Southwestern Medical Center for the gift of DNA samples from a kindred with familial pulmonary fibrosis and Dr. Lisa Boardman from the Mayo Clinic for the colon tumor samples. The longitudinal DNA samples from healthy volunteers were provided by TA Sciences, Inc and some human cell lines were provided by Life Length, Inc. We also thank John Wise for providing the bowhead whale lung cells with permission from the National Marine Sanctuary Foundation. This work was supported by AG01228 from the National Institute on Aging (W.E.W. and J.W.S.), NCI SPORE P50CA70907 (J.W.S.) and the Cancer Prevention Institute of Texas (CPRIT) grants RP1225 (G.D.) and RP160180 (J.W.S.). We also acknowledge the Harold Simmons NCI Designated Comprehensive Cancer Center Support Grant (CA142543), the CPRIT Training Grant RP140110 (T.P.L. and I.M.), and the Southland Financial Corporation Distinguished Chair in Geriatric Research (J.W.S. and W.E.W.). This work was performed in laboratories constructed with support from NIH grant C06 RR30414.

## Author contributions

T.P.L. conducted most of the experiments and wrote the initial draft of the manuscript. J.W.S. and W.E.W. provided supervision and edited the manuscript. N.Z. developed the software for TeSLA quantification under the supervision of G.D. Some of the statistical and mathematical analyses were conducted by J.N. Q-FISH experiments were conducted by I.M. Some of the PBMC experiments and q-PCR quantification were conducted by E. T. E.H. was involved in part of experiments for validation of TeSLA. All authors discussed the results and commented on the manuscript.

## Additional information

**Competing interests:** The authors declare no competing financial interests.

