## [Peer Review File · Nature Communications]

Reviewers' Comments:

Reviewer #1:

Remarks to the Author:

TeSLA is a useful tool that extends beyond the current technical capabilities for telomere analysis. Like several of the existing technologies, TeSLA could be useful across disciplines and will undoubtedly enhance our understanding of telomere biology. The authors have done a thorough analysis of telomeres using TeSLA in the context of aging and cancer, using both cultured cell lines and human samples. In addition, the authors have extended the analysis to different species highlighting the versatility of the assay. The manuscript includes a number of critical controls that highlight the reproducibility of the assay across samples. The results are compelling and the technique appears to improve upon the most sensitive existing assay U-STELA including the development of a platform for automated analysis. However, the technical improvements seem fairly incremental from U-STELA and there are still several limitations including, the degree of technical difficulty and limited throughput. Lastly, although the application of TeSLA and the readership could be broad, the manuscript is written for someone with a bit more expertise (i.e. the average reader won't understand the authors description of U-STELA, especially when it comes to intricate details like the 'panhandle'). In addition, there are a number of poorly constructed sentences and redundancies that make the details of the TeSLA scheme difficult to follow. Overall this was a thorough and critical analysis of a new technique TeSLA, that has the potential to improve the sensitivity and specificity of current telomere analyses.

1. With respect to the design of the TeSLA adapter, the text says that the TeSLA-T adapters have '7 nucleotides of telomeric C-Rich repeats at the 5' end which is complementary to the G-rich overhang..' The primers listed in Supplemental Table 1 show the C-rich sequence at the 3' end, which seems logical given that this region would anneal to the G-rich overhang

2. After enzyme digestion, the fragments are dephosphorylated and this prevents ligation of the genomic DNA fragments with the telomeric fragments. However, in the text the sentence suggests that the ligation is also inhibited by 'adding extra sequences to subtelomeric regions during the next step of TeSLA'. This may be entirely accurate, but that sentence is poorly worded. See full sentence below,

"After enzyme digestion, we performed 5' dephosphorylation using shrimp alkaline phosphatase to prevent ligation between the telomeric DNA fragments and the digested genomic DNA fragments by adding extra sequences to subtelomeric region during the next step of TeSLA."

3. The description of the DNA sequences that are ligated to the genomic DNA and also subtelomeric sequences should be consistent. They are referred to as both 'linkers' and 'adapters' and the supplementary Table 1 calls them 'TeSLA Adapter TA (or AT)'. The description of the adapters in the text is also confusing and redundant,

"The adapters contain phosphorylated 5' AT or TA overhang and a C3 spacer at the 3'end followed by a unique 3' overhang being complementary to adapter primers for the subsequent PCRs. The adapters were designed to contain phosphorylated 5' AT or TA overhang and a C3 spacer at the 3' end followed by a unique 3' overhang being complementary to adapter primers to facilitate ligation to only occur between the 5' end of adapters and 3' end of genomic/telomeric C-rich DNA fragments."

4. Figure 1a should be described succinctly, but with a bit more detail in the legend. The use of 'Spacer C3' could also be better described and denoted in the figure. The supplemental table describes that both the TeSLA adapter short and the TeSLA adapter TA or AT have Spacer C3 at the 3' termini, but the depiction of this region in the schematic is confusing as the Spacer C3 only appears on one strand of the double-strand adaptor?

5. The experiments outlined in Figure 4F demonstrate the reproducibility of the technique across the same sample, but how consistent is this from user to user. There is no internal control to ensure efficient amplification across all reactions.
6. The comparisons in Figures 2 and 6 clearly demonstrate a benefit to using TeSLA over TRF assays, but TRF assays are certainly less sensitive than U-STELA. How would U-STELA compare to TeSLA in these same assays?
7. The AVG TL (Kb), Shortest 20% (Kb), and <1.6kb (%) for the Imetelstat data should be included in Figure 2C.

In the materials and methods, the southern blot protocol should be detailed and not just referenced given that this is really the basis for the image acquisition and analysis for TeSLA.

Reviewer #2:

Remarks to the Author:

The group of Shay and Wright report on an exceedingly useful new method to detect the shortest telomeres in cells, i.e. those telomeres that are the most significant with regard to the effect of telomeres on cell proliferation. This method, called TeSLA, is a major advantage over prior methods. Compared to Yp/Xp STELA, TeSLA has the advantage of detecting all chromosome ends and compared to Universal STELA (U-STELA), the new method is much more robust, has lower background, and could be used for non-human mammals. This new method will be welcomed by the community and will hopefully herald in a new era where telomere length studies don't merely measure the amount of TTAGGG repeats in cells but report the frequency of the most relevant telomeres, the shortest ones.

Despite these laudatory remarks, there are some aspects of this study that hopefully can be improved in a revised manuscript. The major disadvantage of this technique is that in cells with very long telomeres (above 15 kb), the majority of the telomeres escape detection. This makes it difficult to evaluate change in the frequency of short telomeres since the denominator is missing. In particular, this problem applies to human cells with very long telomeres (e.g. ALT cells) and to non-human mammals (e.g. mice). Below I suggest methods to allow the simple band counting to be informative, even in such samples. The authors may have their own ideas on how to do this. Addition of whatever method they deem most suitable would make the paper more valuable.

Figure 4 a and b;

Fig 4b shows that average TL in U2OS is 3.48 kb which is quite a bit shorter than that of measured by TRF analysis. This reviewer agrees that TRF analysis may overestimate the TL, however, TeSLA may underestimate telomere length in ALT cells. First, TeSLA has upper limit of telomere detection as 16-18kb. Second, Nabetani and Ishikawa (2009) showed that both G and C strand telomeres in ALT cells contain large numbers of internal gaps and/or nicks. TeSLA PCR can presumably prime at the telomere internal C strand gaps and results in shorter PCR fragments. To test this possibility, TeSLA PCR need to be performed using telomeric DNA of different size classes (e.g. >10 kb, 10-5 kb, <5 kb) extracted from agarose gel. If the PCR starts at telomere internal gap/nick the products will be smaller than expected from the size class isolated. If, however, the >10 kb DNA yields only large products (10 to the 16 kb cut-off), this reviewer's worry is unfounded.

Figure 6a and b

I recommend that author present data from 15 individuals in dot blot format to preserve details of each data points. Averaging base line sample length from 15 individuals will lead to a large spread in the data and ditto for the 1 year data points. Better would be to measure the kb change for

each individual and plot that. It is also not clear how the p values are calculated in the current format of the data representation.

Supplemental Fig 4a shows that telomere shortening by 0.28 kb per year occurs in PBMCs. Is the value an overestimate? It is much higher than previously reported.

Figure 7a

It would be extremely valuable if TeSLA is useful for the analysis of mouse telomeres since so much of telomere biology is studied in this model organism. However, since TeSLA does not contain internal control and most of the mouse telomeres are above the upper limit of amplification, the results cannot be normalized using longer fragments vs shorter fragments. Variability of amount genomic DNA loaded and ligation efficiency between samples may influence the results. The authors should come up with a method to make simple band counting of the detectable telomeres a reliable estimate of changes in telomere length. One method would be to show three independent DNA preps of the same genotype/age and show that such independent TeSLA experiments (independent cutting, ligation, PCR etc) yield the same number of bands. It also would be helpful if some internal PCR trick could be found to evaluate equal loading. Although most of the genome will be cut into very small fragments, there may be a locus that can be found. Otherwise, the author should explicitly indicate how the amount of input DNA was measured. The authors may be able to think of another method to deal with the 'denominator' problem noted in the general remarks. In any case, addition of a way to make the analysis of mouse telomeres solid would be key.

Related: Did the authors try polymerases other than FialSafe polymerase kit to improve PCR products in the higher MW range?

Minor points

Figure 4d; Author describes "using 8 reactions is reasonable to estimate the mean TL."

This reviewer is not sure 8 reactions means 8 PCR reactions using one ligated sample or 8 PCR using 8 independently processed samples.

Page 8; TeSLA adapter is referred as "linkers". For consistency, use "adapters" instead of "linkers".

Page 8; The sentence "The adapters contain phosphorylated 5' AT or TA overhang and a C3 spacer at the 3' end followed by a unique 3' overhang being complementary to adapter primers for the subsequent PCRs." is repeated.

Discussion: "Until the development of TeSLA, there was no technique capable of quantitating the entire spectrum of telomeres in ALT cells." Considering the upper detection limit of TL, "entire spectrum" is an overstatement.

Abstract: The first sentence of the Abstract does not make sense to this reviewer. Telomere length does not induce the DDR. Perhaps: Replicative senescence is triggered when the shortest telomeres in a cell lose the ability to protect chromosomes ends.

Page 9, at the bottom: 'more short telomeres' higher frequency of short telomeres? Or just shorter telomeres?

Reviewer #3:

Remarks to the Author:

Lai et al describe a new technique (TeSLA; Telomere Shortest Length Assay), for measuring the distribution of the shortest telomeres in a cell population. There are currently several techniques used to measure telomere length. Most notably, TRF analysis and Flow-FISH have emerged as the gold standard techniques, whilst qPCR is used most widely, but is renowned for being variable and error prone. One of the major limitations of many of the currently used telomere length tests, is the measurement of relative telomere content. This is relevant, but provides limited information about the shortest telomeres, which are the responsible for DNA damage response activation and cellular senescence. TeSLA overcomes these problems by amplifying individual telomeres to provide individual telomere lengths. TeSLA is an improvement on STELA (single telomere length analysis), which is only able to analyse the telomeres from specific chromosome ends (due to limitations in suitable primer sites at all chromosome ends). TeSLA appears to be very similar to universal-STELA, with some obvious improvements in the adapter sequences/PCR amplification and the restriction enzyme cocktail (although both MseI and NdeI are used in universal-STELA). The more significant development that this manuscript provides is the automation of image processing, band detection and annotation, and information output, which has been done using Matlab. Overall, the experimentation is very well executed and thorough. The TeSLA gels are beautiful! This is, however, a complex and committed experimental technique. Both STELA and universal-STELA are only carried out in a limited number of labs, and I question how readily this technique can be applied by non-specialised labs. It is also unclear how much of an advance TeSLA is over universal-STELA.

Points to address:

- (i) Does TeSLA amplify extrachromosomal telomeric sequences? Have you been able to control for this?
- (ii) In the overview of current telomere length quantitation approaches, it would be useful to mention telomere length measurement tools that are applied to WGS (eg TelSeq, Computel, qMotif, TelomereHunter). These are gaining traction in genomics studies.
- (iii) The TRF in Fig 2 is of poor quality and should be improved.
- (iv) In Supp Fig 2b, were equal amounts of DNA or equal cell numbers loaded? How was this normalised?
- (v) I am struggling lining up the band intensity profile (Fig 3d) with the bands on the TeSLA image (Fig 3b). Could the authors mark the specific bands (maybe with different coloured dots).
- (vi) What is the reason for the variable band intensities on the TeSLA gels? How well does the software pick up the weak bands? Does the software detect the very small bands that are present in the bowhead whale?
- (vii) The TRFs are all run by standard gel electrophoresis, whereas pulsed field gel electrophoresis (PFGE) is routinely used and provides much greater resolution. How do TRFs run by PFGE compare (Fig 6)?
- (viii) It would be useful to provide a couple of sentences describing how the fragments are separated and visualised in the manuscript. This information is lacking from the main text, but would help explain the technique.
- (ix) Was TeSLA performed on 3 separate blood samples from the 32-year old and 72-year old donors, or were they 3 different experiments on the same DNA (this is unclear from the text).
- (x) Can some statistics be provided for the telomere length changes observed by TeSLA (eg for the Imetelstat experiment, for the 32-year old/72-year old experiment and for the colon cancer experiment).
- (xi) How were the length and median etc calculations made for the TRF (Fig 6)?

Reviewer #1 (Remarks to the Author):

TeSLA is a useful tool that extends beyond the current technical capabilities for telomere analysis. Like several of the existing technologies, TeSLA could be useful across disciplines and will undoubtedly enhance our understanding of telomere biology. The authors have done a thorough analysis of telomeres using TeSLA in the context of aging and cancer, using both cultured cell lines and human samples. In addition, the authors have extended the analysis to different species highlighting the versatility of the assay. The manuscript includes a number of critical controls that highlight the reproducibility of the assay across samples. The results are compelling and the technique appears to improve upon the most sensitive existing assay U-STELA including the development of a platform for automated analysis. However, the technical improvements seem fairly incremental from U-STELA and there are still several limitations including, the degree of technical difficulty and limited throughput. Lastly, although the application of TeSLA and the readership could be broad, the manuscript is written for someone with a bit more expertise (i.e. the average reader won't understand the authors description of U-STELA, especially when it comes to intricate details like the 'panhandle'). In addition, there are a number of poorly constructed sentences and redundancies that make the details of the TeSLA scheme difficult to follow. Overall this was a thorough and critical analysis of a new technique TeSLA, that has the potential to improve the sensitivity and specificity of current telomere analyses.

First, we would like to thank the reviewer for recognizing that the assay we have developed has “the potential to improve the sensitivity and specificity of current telomere analyses” and that the application of this technique could be broad. We also want to apologize for some poor writing and the use of uncommon terminology. We have completely addressed these in the revised manuscript.

1. With respect to the design of the TeSLA adapter, the text says that the TeSLA-T adapters have ‘7 nucleotides of telomeric C-Rich repeats at the 5’ end which is complementary to the G-rich overhang. The primers listed in Supplemental Table 1

show the C-rich sequence at the 3' end, which seems logical given that this region would anneal to the G-rich overhang

A: We have corrected the sentence to be “Each TeSLA-T contains 7 nucleotides of telomeric C-rich repeats at the 3' end which is complementary to the G-rich overhang followed by a unique sequence derived from bacteriophage MS2 for PCR”.

2. After enzyme digestion, the fragments are dephosphorylated and this prevents ligation of the genomic DNA fragments with the telomeric fragments. However, in the text the sentence suggests that the ligation is also inhibited by ‘adding extra sequences to subtelomeric regions during the next step of TeSLA’. This may be entirely accurate, but that sentence is poorly worded. See full sentence below,

“After enzyme digestion, we performed 5' dephosphorylation using shrimp alkaline phosphatase to prevent ligation between the telomeric DNA fragments and the digested genomic DNA fragments by adding extra sequences to subtelomeric region during the next step of TeSLA.”

A: We have corrected this and appreciate how our wording could have been misinterpreted. The new statement is as follows. “After RE digestion, we performed 5' dephosphorylation using shrimp alkaline phosphatase to prevent non-specific ligation between the telomeric DNA fragments and the digested genomic DNA fragments which could potentially add extra sequences to subtelomeric region during the next step of TeSLA for adapter ligation.

3. The description of the DNA sequences that are ligated to the genomic DNA and also subtelomeric sequences should be consistent. They are referred to as both ‘linkers’ and ‘adapters’ and the supplementary Table 1 calls them ‘TeSLA Adapter TA (or AT)’. The description of the adapters in the text is also confusing and redundant,

“The adapters contain phosphorylated 5' AT or TA overhang and a C3 spacer at the 3' end followed by a unique 3' overhang being complementary to adapter primers for the

subsequent PCRs. The adapters were designed to contain phosphorylated 5' AT or TA overhang and a C3 spacer at the 3' end followed by a unique 3' overhang being complementary to adapter primers to facilitate ligation to only occur between the 5' end of adapters and 3' end of genomic/telomeric C-rich DNA fragments."

A: We agree and have revised "linkers" to be termed "adapters".

4. Figure 1a should be described succinctly, but with a bit more detail in the legend. The use of 'Spacer C3' could also be better described and denoted in the figure. The supplemental table describes that both the TeSLA adapter short and the TeSLA adapter TA or AT have Spacer C3 at the 3' termini, but the depiction of this region in the schematic is confusing as the Spacer C3 only appears on one strand of the double-strand adaptor?

A: We have detailed the legend of Figure 1a and added the depiction of Spacer C3 to the 3' end of TeSLA adapter short in Figure 1a.

5. The experiments outlined in Figure 4F demonstrate the reproducibility of the technique across the same sample, but how consistent is this from user to user. There is no internal control to ensure efficient amplification across all reactions.

A: We have shared our TeSLA to other two labs. Both labs have indicated that they can perform TeSLA without any technical problems. We include in the revised text that other labs have beta tested this method without technical problems. We can include the names of these labs with whom we have shared this method with if appropriate and requested.

6. The comparisons in Figures 2 and 6 clearly demonstrate a benefit to using TeSLA over TRF assays, but TRF assays are certainly less sensitive than U-STELA. How would U-STELA compare to TeSLA in these same assays?

A: We have demonstrated that TeSLA is more sensitive and specific than U-STELA for telomere detection in Figure 1b and c.

7. The AVG TL (Kb), Shortest 20% (Kb), and <1.6kb (%) for the lmetelstat data should be included in Figure 2C.

A: We have added quantification results in Figure 2c.

In the materials and methods, the southern blot protocol should be detailed and not just referenced given that this is really the basis for the image acquisition and analysis for TeSLA.

A: We agree and have detailed the Southern blot protocol in more detail in the revised manuscript.

Reviewer #2 (Remarks to the Author):

The group of Shay and Wright report on an exceedingly useful new method to detect the shortest telomeres in cells, i.e. those telomeres that are the most significant with regard to the effect of telomeres on cell proliferation. This method, called TeSLA, is a major advantage over prior methods. Compared to Yp/Xp STELA, TeSLA has the advantage of detecting all chromosome ends and compared to Universal STELA (U-STELEA), the new method is much more robust, has lower background, and could be used for non-human mammals. This new method will be welcomed by the community and will hopefully herald in a new era where telomere length studies don't merely measure the amount of TTAGGG repeats in cells but report the frequency of the most relevant telomeres, the shortest ones.

We thank the reviewer for recognizing how important this new method is and we have fully addressed the concerns raised by this reviewer in the revised manuscript.

Despite these laudatory remarks, there are some aspects of this study that hopefully can be improved in a revised manuscript. The major disadvantage of this technique is that in cells with very long telomeres (above 15 kb), the majority of the telomeres escape detection. This makes it difficult to evaluate change in the frequency of short telomeres since the denominator is missing. In particular, this problem applies to human

cells with very long telomeres (e.g. ALT cells) and to non-human mammals (e.g. mice). Below I suggest methods to allow the simple band counting to be informative, even in such samples. The authors may have their own ideas on how to do this. Addition of whatever method they deem most suitable would make the paper more valuable.

We truly thank the reviewer for spending the time to make valuable suggestions that we address below. Our only goal is to have a robust method that will be adopted and used by the telomere community.

Figure 4 a and b;

Fig 4b shows that average TL in U2OS is 3.48 kb which is quite a bit shorter than that of measured by TRF analysis. This reviewer agrees that TRF analysis may overestimate the TL, however, TeSLA may underestimate telomere length in ALT cells. First, TeSLA has upper limit of telomere detection as 16-18kb. Second, Nabetani and Ishikawa (2009) showed that both G and C strand telomeres in ALT cells contain large numbers of internal gaps and/or nicks. TeSLA PCR can presumably prime at the telomere internal C strand gaps and results in shorter PCR fragments. To test this possibility, TeSLA PCR need to be performed using telomeric DNA of different size classes (e.g. >10 kb, 10-5 kb, <5 kb) extracted from agarose gel . If the PCR starts at telomere internal gap/nick the products will be smaller than expected from the size class isolated. If, however, the >10 kb DNA yields only large products (10 to the 16 kb cut-off), this reviewer's worry is unfounded.

A: We agree that we might underestimate TLs in U2OS cells. We also performed the experiments the reviewer suggested (see below). We found that TeSLA might not be able to measure accurately TLs in ALT cells. Thus, we removed data from U2OS cells and used HeLa LT (telomerase positive and with long telomeres) to detect the upper size limit of TL detection by TeSLA. Our results show that the upper size limit for TL detection by TeSLA is ~18 kb which is consistent with the results using U2OS cells. We also evaluated intra-variation of TeSLA using normal human bronchial epithelial cells (HBEC). We found consistent results for intra-variation of TeSLA using U2OS and HBEC cells.

TeSLA results of U2OS with different size classes.

After TeSLA ligation, ligated DNA was separated with 1 % agarose and then extracted from agarose gel with different size classes (>10 kb, 10-5 kb, and <5kb). + indicates ligated U2OS DNA without size selection.

Figure 6a and b

I recommend that author present data from 15 individuals in dot blot format to preserve details of each data points. Averaging base line sample length from 15 individuals will lead to a large spread in the data and ditto for the 1 year data points. Better would be to

measure the kb change for each individual and plot that. It is also not clear how the p values are calculated in the current format of the data representation.

A: We have changed this figure to indicate the kb change and p values of each.

Supplemental Fig 4a shows that telomere shortening by 0.28 kb per year occurs in PBMCs. Is the value an overestimate? It is much higher than previously reported.

A: We have mentioned this observation in the discussion. We believe that we do not overestimate telomere shortening per year in PBMCs using TeSLA. TeSLA is very sensitive for detecting the shortest telomeres in a heterogeneous telomere background. However TRF analysis and q-PCR only measure average telomere length from PBMCs without measuring the shortest telomeres. TeSLA is capable of measure sub-populations of cells in PBMCs, such as CD28- T cells, that have shorter telomeres and a higher TL shortening rate compared to other sub-types of cells in PBMCs.

Figure 7a

It would be extremely valuable if TeSLA is useful for the analysis of mouse telomeres since so much of telomere biology is studied in this model organism. However, since TeSLA does not contain internal control and most of the mouse telomeres are above the upper limit of amplification, the results cannot be normalized using longer fragments vs shorter fragments. Variability of amount genomic DNA loaded and ligation efficiency between samples may influence the results. The authors should come up with a method to make simple band counting of the detectable telomeres a reliable estimate of changes in telomere length. One method would be to show three independent DNA preps of the same genotype/age and show that such independent TeSLA experiments (independent cutting, ligation, PCR etc) yield the same number of bands. It also would be helpful if some internal PCR trick could be found to evaluate equal loading. Although most of the genome will be cut into very small fragments, there may be a locus that can be found. Otherwise, the author should explicitly indicate how the amount of input DNA was measured. The authors may be able to think of another

method to deal with the 'denominator' problem noted in the general remarks. In any case, addition of a way to make the analysis of mouse telomeres solid would be key?

A: We have used 3 different DNA preps from mouse liver (same mouse) to perform independent TeSLA. We used q-PCR to quantify DNA input before we performed TeSLA and then used the same amount (50 ng of genomic DNA) to perform TeSLA. We found all 3 independent TeSLAs from TERT KO G4 have more amplified telomeres than TERT KO heterozygous mice.

Minor points

Figure 4d; Author describes "using 8 reactions is reasonable to estimate the mean TL." This reviewer is not sure 8 reactions means 8 PCR reactions using one ligated sample or 8 PCR using 8 independently processed samples.

A: We have changed the text to be 8 PCR reactions

Page 8; TeSLA adapter is referred as "linkers". For consistency, use "adapters" instead of "linkers".

A: For consistency, we have changed "linkers" to be "adapters" throughout the revised manuscript.

Page 8; The sentence "The adapters contain phosphorylated 5' AT or TA overhang and a C3 spacer at the 3' end followed by a unique 3' overhang being complementary to adapter primers for the subsequent PCRs." is repeated.

A: We apologize for this oversight. In the revised manuscript we have deleted the repeated sentence.

Discussion: "Until the development of TeSLA, there was no technique capable of quantitating the entire spectrum of telomeres in ALT cells." Considering the upper detection limit of TL, "entire spectrum" is an overstatement.

A: We agree and have eliminated the term "entire spectrum" in the revised text. However, even in ALT cells it is believed that it is the shortest telomeres that are important in initiating DNA recombination. It is unclear at the present if and why

knowledge of the longest telomeres in ALT cells is of biological importance.

Abstract: The first sentence of the Abstract does not make sense to this reviewer. Telomere length does not induce the DDR. Perhaps: Replicative senescence is triggered when the shortest telomeres in a cell lose the ability to protect chromosome ends.

A: We agree and have revised this poorly constructed statement.

Page 9, at the bottom: 'more short telomeres' higher frequency of short telomeres? Or just shorter telomeres?

A: We have revised the text to be more accurate in the revised text

Reviewer #3 (Remarks to the Author):

Lai et al describe a new technique (TeSLA; Telomere Shortest Length Assay), for measuring the distribution of the shortest telomeres in a cell population. There are currently several techniques used to measure telomere length. Most notably, TRF analysis and Flow-FISH have emerged as the gold standard techniques, whilst qPCR is used most widely, but is renowned for being variable and error prone. One of the major limitations of many of the currently used telomere length tests, is the measurement of relative telomere content. This is relevant, but provides limited information about the shortest telomeres, which are the responsible for DNA damage response activation and cellular senescence. TeSLA overcomes these problems by amplifying individual telomeres to provide individual telomere lengths. TeSLA is an improvement on STELA (single telomere length analysis), which is only able to analyse the telomeres from specific chromosome ends (due to limitations in suitable primer sites at all chromosome ends). TeSLA appears to be very similar to universal-STELA, with some obvious improvements in the adapter sequences/PCR amplification and the restriction enzyme cocktail (although both MseI and NdeI are used in universal-STELA). The more significant development that this manuscript provides is the automation of

image processing, band detection and annotation, and information output, which has been done using Matlab. Overall, the experimentation is very well executed and thorough. The TeSLA gels are beautiful! This is, however, a complex and committed experimental technique. Both STELA and universal-STELA are only carried out in a limited number of labs, and I question how readily this technique can be applied by non-specialised labs. It is also unclear how much of an advance TeSLA is over universal-STELA.

A: We would like to thank the reviews for indicating that the experiments were very well executed and thorough and indicating that the TeSLA gels are beautiful. We recognize and fully appreciate the concerns of this reviewer and have fully addressed them in the revised manuscript.

Points to address:

(i) Does TeSLA amplify extrachromosomal telomeric sequences? Have you been able to control for this?

A: TRF analysis, Q-PCR, and other methods for telomere length detection are not able to control the detection of extrachromosomal telomeric sequences. It is still not clear if extrachromosomal telomeric sequences contain subtelomeric DNA sequences or only telomeric DNA repeats. Thus, we cannot rule out the possibility of amplifying extrachromosomal telomeric sequences by TeSLA. However, if there is only double-stranded telomeric DNA in extrachromosomal telomeric sequences, the TeSLA method would not be able to detect these extrachromosomal telomeric sequences. We have included a statement indicating some of the limitations of all telomere length detection methods including TeSLA.

(ii) In the overview of current telomere length quantitation approaches, it would be useful to mention telomere length measurement tools that are applied to WGS (eg TelSeq, Computel, qMotif, TelomereHunter). These are gaining traction in genomics studies.

A: We have mentioned these techniques in the introduction section of the revised text.

(iii) The TRF in Fig 2 is of poor quality and should be improved.

A: Since TRF analysis is not able to detect the shortest telomeres due to probe hybridization kinetics, we increased the intensity of this TRF gel to indicate that some shorter telomeres can be detected but not those below 0.8kb.

(iv) In Supp Fig 2b, were equal amounts of DNA or equal cell numbers loaded? How was this normalised?

A: In Supplementary Fig 2b, we used equal amounts of DNA that quantified using Nanodrop. This has been clearly indicated in the revised text.

(v) I am struggling lining up the band intensity profile (Fig 3d) with the bands on the TeSLA image (Fig 3b). Could the authors mark the specific bands (maybe with different coloured dots).

A: We have changed Fig 3b with a gel image on top of the plot.

(vi) What is the reason for the variable band intensities on the TeSLA gels? How well does the software pick up the weak bands? Does the software detect the very small bands that are present in the bowhead whale?

A: It depends on the PCR efficiency and probe hybridization. Short sequences tend to have light bands intensity. Sometimes two or more bands overlap with each other, which will also increase the intensity value. Our software is designed to identify the overlapping bands that show more intense staining.

The software first identifies the centers of lanes and then plots the intensity profile for each individual lane. 1D watershed segmentation is applied to detect significant peaks on the lane intensity profile, which indicate the centers of bands. There is a band threshold parameter, which can be adjusted by the user, to distinguish real bands from the background noise. Thus, if a band is approximately twice as intense as adjacent bands, the software will calculate this as two telomeres that overlap. The lower the parameter is, the more sensitive the software will be on weak bands detection. But we

recommend users to use consistent threshold parameters for the entire project. The software developed also permit manual adjustment options for users to add or delete bands from the software detection results. More details are provided in the revised text.

(vii) The TRFs are all run by standard gel electrophoresis, whereas pulsed field gel electrophoresis (PFGE) is routinely used and provides much greater resolution. How do TRFs run by PFGE compare (Fig 6)?

A: The pulsed field gel electrophoresis is for separation of large DNA molecules (~15-20 kb or larger). Since most normal human telomeres are less than 10 kb, we do not think it is necessary to do TRF analysis using PFGE even though for mouse studies and perhaps ALT cells this would have utility.

(viii) It would be useful to provide a couple of sentences describing how the fragments are separated and visualised in the manuscript. This information is lacking from the main text, but would help explain the technique.

A: We have provided descriptions of these steps in our revised manuscript.

(ix) Was TeSLA performed on 3 separate blood samples from the 32-year old and 72-year old donors, or were they 3 different experiments on the same DNA (this is unclear from the text).

A: We used the same DNA to do experiments on different days (independent ligations, restriction enzyme digestion, PCR, and telomere detection). This information is now included in the revised text.

(x) Can some statistics be provided for the telomere length changes observed by TeSLA (eg for the Imetelstat experiment, for the 32-year old/72-year old experiment and for the colon cancer experiment).

A: We have provided quantification data for every TeSLA result in the revised figures.

(xi) How were the length and median etc calculations made for the TRF (Fig 6)?

A: We have used Image Quant software to quantify telomere lengths for our TRF analysis as described ¹

1. Mender I, Shay JW. Telomere Restriction Fragment (TRF) Analysis. *Bio Protoc* **5**, (2015).

Reviewers' Comments:

Reviewer #1:

Remarks to the Author:

Lai et al. have made revisions to their manuscript "TeSLA: A Method for Measuring the Distribution of the Shortest Telomeres in Cells and Tissues". Overall, I found the authors to be responsive to the reviewer's comments and critiques and the revisions have strengthened the manuscript. TeSLA will unquestionably be a useful tool in telomere length analysis and will undoubtedly enhance our understanding of telomere biology. This manuscript will be a nice contribution to the field.

Reviewer #2:

Remarks to the Author:

Overall, this reviewer is satisfied by the author's responses in revised manuscript, however, there are a few minor points need to be addressed.

1. A part of original remark from this reviewer about Figure 7a is missing from author's response. "Figure 7a It would be extremely valuable if TeSLA is useful for... etc.

Missing: Related: Did the authors try polymerases other than FailSafe polymerase kit to improve PCR products in the higher MW range?"

Could the authors comment on this in the text?

2. q-PCR quantification of DNA

I suppose that the data in Fig 6b is from the q-PCR of mouse B1 repeat. This is not clearly indicated neither in Fig. 6b nor in the legend. And the meaning of the y-axis is unclear. Please clarify in the legend. Please also add a brief description about of q-PCR method and more detail on quantification and qualitative analysis of genomic DNA in Methods.

3. U2OS

Since the results with U2OS ALT cells (Fig. 4b in original manuscript) has been omitted from the revised manuscript, reference to "ALT (alternative lengthening of telomere) cell lines" should be deleted. And in the Methods section, HBEC should replace U2OS.

4. Supplemental Figure 3C

Please clarify in the legend what the meaning is of the small and large circles in the figure.

Reviewer #3:

Remarks to the Author:

The authors have adequately addressed all my concerns. This includes textual changes, improving the clarity of figures, and expanding and clarifying technical details.

Reviewer #1 (Remarks to the Author):

Lai et al. have made revisions to their manuscript “TeSLA: A Method for Measuring the Distribution of the Shortest Telomeres in Cells and Tissues”. Overall, I found the authors to be responsive to the reviewer’s comments and critiques and the revisions have strengthened the manuscript. TeSLA will unquestionably be a useful tool in telomere length analysis and will undoubtedly enhance our understanding of telomere biology. This manuscript will be a nice contribution to the field.

Reviewer #2 (Remarks to the Author):

Overall, this reviewer is satisfied by the author’s responses in revised manuscript, however, there are a few minor points need to be addressed.

1. A part of original remark from this reviewer about Figure 7a is missing from author’s response.

“Figure 7a It would be extremely valuable if TeSLA is useful for... etc.

Missing: Related: Did the authors try polymerases other than FailSafe polymerase kit to improve PCR products in the higher MW range?”

Could the authors comment on this in the text?

A: We have addressed this in our revised manuscript. We did not use other PCR enzymes other than FailSafe polymerase. Although it is possible that other PCR enzymes are also suitable for TeSLA PCR, Failsafe polymerase has been used to reliably perform long-range PCR and amplify telomeric DNA.

2. q-PCR quantification of DNA

I suppose that the data in Fig 6b is from the q-PCR of mouse B1 repeat. This is not clearly indicated neither in Fig. 6b nor in the legend. And the meaning of the y-axis is unclear. Please clarify in the legend. Please also add a brief description about of q-PCR method and more detail on quantification and qualitative analysis of genomic DNA in Methods.

A: We have added q-PCR quantification to the section of methods and in the legend for supplementary figure 6.

3. U2OS

Since the results with U2OS ALT cells (Fig. 4b in original manuscript) has been omitted from the revised manuscript, reference to “ALT (alternative lengthening of telomere) cell lines” should be deleted. And in the Methods section, HBEC should replace U2OS.

A: We have replaced U2OS to be HBEC in the Methods section.

4. Supplemental Figure 3C

Please clarify in the legend what the meaning is of the small and large circles in the figure.

A: We have described the large circles in Supplementary Fig. 3C. “The scatter plot represents distributions of TLs from TeSLA results (16 reactions) of HeLa LT cells. Each circle represents a particular TL that was detected by TeSLA. The circle size indicates single (small circle) or multiple (large circle) counts for a particular TL.”

Reviewer #3 (Remarks to the Author):

The authors have adequately addressed all my concerns. This includes textual changes, improving the clarity of figures, and expanding and clarifying technical details.